# Frequency modulation of ERK activation dynamics rewires cell fate

Hyunryul Ryu[1,2], Minhwan Chung[1], Maciej Dobrzyński[3], Dirk Fey[3], Yannick Blum[4], Sung Sik Lee[5], Matthias Peter[5], Boris N Kholodenko[3,*], Noo Li Jeon[1,2,**] & Olivier Pertz[4,†,***]

## Abstract

Transient versus sustained ERK MAP kinase (MAPK) activation dynamics induce proliferation versus differentiation in response to epidermal (EGF) or nerve (NGF) growth factors in PC-12 cells. Duration of ERK activation has therefore been proposed to specify cell fate decisions. Using a biosensor to measure ERK activation dynamics in single living cells reveals that sustained EGF/NGF application leads to a heterogeneous mix of transient and sustained ERK activation dynamics in distinct cells of the population, different than the population average. EGF biases toward transient, while NGF biases toward sustained ERK activation responses. In contrast, pulsed growth factor application can repeatedly and homogeneously trigger ERK activity transients across the cell population. These datasets enable mathematical modeling to reveal salient features inherent to the MAPK network. Ultimately, this predicts pulsed growth factor stimulation regimes that can bypass the typical feedback activation to rewire the system toward cell differentiation irrespective of growth factor identity.

Keywords  cell fate decisions; ERK activity dynamics; FRET biosensor; single cell biology; signaling heterogeneity
Subject Categories  Quantitative Biology & Dynamical Systems; Signal Transduction; Development & Differentiation
Mol Syst Biol. (2015) 11: 838

See also: N Blüthgen (November 2015)

## Introduction

Complex signaling networks allow cells to translate external stimuli into specific cell fates. In many cases, signaling dynamics rather than steady states control these fate decisions (Levine *et al*, 2013; Purvis & Lahav, 2013). Furthermore, signaling states of individual cells differ even across an isogenic population (Cohen-Saidon *et al*, 2009; Snijder & Pelkmans, 2011; Chen *et al*, 2012), due to broad distributions of protein abundances, as well as intrinsic noise present within all biochemical networks (Snijder & Pelkmans, 2011). Measuring single-cell signaling dynamics is therefore a key to understand how cellular responses correlate with fate decisions.

The extracellular-regulated kinase (ERK) regulates cellular fates such as proliferation and differentiation. It functions within a mitogen-activated protein kinase (MAPK) signaling network in which growth factor (GF) receptors activate a Ras GTPase, subsequently triggering a MAPK cascade leading to ERK activation (Avraham & Yarden, 2011). Rat adrenal pheochromocytoma PC-12 cells have been used as a model system to study MAPK-dependent fate decisions (Marshall, 1995). Stimulation with EGF or NGF leads to transient or sustained ERK activation dynamics, respectively, triggering proliferation or differentiation (Marshall, 1995; Avraham & Yarden, 2011). These different ERK activation dynamics involve activation of different Ras isoforms (Sasagawa *et al*, 2005), as well as GF-dependent control of the MAPK network topology (Santos *et al*, 2007), with negative or positive feedbacks producing adaptive or bistable outputs (Xiong & Ferrell, 2003; Santos *et al*, 2007; Avraham & Yarden, 2011). Downstream, molecular interpretation of signal duration involves stabilization of ERK-induced immediate early gene (IEG) products by sustained ERK activity, ultimately instructing the differentiation fate (Murphy *et al*, 2002; Nakakuki *et al*, 2010). Single-cell analysis has, however, revealed that NGF does not lead to homogeneous PC-12 cell differentiation. Rather, a heterogeneous mix of differentiating and proliferating cells is observed, with the respective cell fate choices depending on a complex ERK and AKT signaling code (Chen *et al*, 2012). Here, we study ERK activation dynamics in GF-stimulated single PC-12 cells. We find that sustained GF stimulation induces heterogeneous cell responses different than the population average, with both GFs being able to produce transient and sustained ERK activation responses. We dynamically probe the ERK signaling flux through application of GF pulses, which homogenizes ERK activation

1  School of Mechanical and Aerospace Engineering, Seoul National University, Seoul, Korea
2  Institute of Advanced Machinery and Design, Seoul National University, Seoul, Korea
3  System Biology Ireland, University College Dublin, Belfield, Dublin, Ireland
4  Department of Biomedicine, University of Basel, Basel, Switzerland
5  Institute of Biochemistry, Zurich, Switzerland
   *Corresponding author. Tel: +353 1716 6331; E-mail: boris.kholodenko@ucd.ie
   **Corresponding author. Tel: +82 2 880 71 11; E-mail: njeon@snu.ac.kr
   ***Corresponding author. Tel: +41 31 631 46 16; E-mail: olivier.pertz@izb.unibe.ch
   †Present address: Institute of Cell Biology, University of Bern, Bern, Switzerland

responses throughout the cell population. This provides novel insight to understand the MAPK network structure and ultimately provides a rationale to rewire cell fate decisions independently of GF identity.

# Results

### Sustained GF stimulation induces heterogeneous ERK activation dynamics

To study ERK activation dynamics in single PC-12 cells, we produced a stable cell line that expresses EKAR2G, a fluorescence resonance energy transfer-based biosensor for endogenous ERK activity (Fig 1A) (Harvey *et al*, 2008; Fritz *et al*, 2013). This biosensor specifically reports on ERK, but not on p38 mitogen-activated, neither on c-Jun N-terminal kinases (Harvey *et al*, 2008). By virtue of a nuclear export sequence, EKAR2G localizes to, and specifically measures ERK activity in the cytosol (Fig 1B). Although this does not seem to be true for all cell types (Ahmed *et al*, 2014), we assumed that cytosolic and nuclear pools of ERK activity are in equilibrium, since, at least for EGF-stimulated PC-12 cells, there is no apparent time lag between nuclear and cytosolic ERK activation dynamics (Herbst *et al*, 2011). Biosensor expression levels were homogeneous and displayed only small standard deviation with respect to the population median (Appendix Fig S1A). To match the temporal resolution enabled by our biosensor, we used a flow-based, computer-programmable microfluidic device to deliver GFs with precise kinetics (Fig 1C). We observed that flow induced transient ERK activation (Appendix Fig S1B and C). This most likely results from flow-induced mechanical stress, and/or exposure to low amount of serum required for cell survival in the live cell imaging experiments. Subsequent experiments were then performed after the flow effect has subsided, leading to a flat baseline (Appendix Fig S1D).

Throughout this study, we used high-dosage, 25 ng/ml EGF and 50 ng/ml NGF (representing equivalent GF molarities (Santos *et al*, 2007)), and low-dosage, 1 ng/ml EGF and 2 ng/ml NGF concentrations. As previously described (Santos *et al*, 2007), Western blot analysis showed that high-dosage EGF triggered a single ERK activation peak that almost nearly returned to baseline after 10–15 min (Fig 1D). In contrast, high-dosage NGF evoked one ERK activation peak followed by sustained, but reduced with respect to the peak, ERK activity. Similar ERK activation dynamics profiles were observed in population-averaged EKAR2G measurements (Fig 1E). At the single-cell level, however, a mix of transient, oscillatory, and sustained ERK activity trajectories were observed (Fig 1F–H, Video EV1), showing that EGF- and NGF-induced ERK activation kinetics are the result of heterogeneous cell responses distinct from the population average. This signaling heterogeneity was not a consequence of the small but existing heterogeneity in EKAR2G expression levels. Specifically, we have not observed any correlation between biosensor expression levels and NGF-triggered ERK activity at 40′ after stimulation, a time point and experimental condition at which highly heterogeneous ERK activity is observed in the cell population (Appendix Fig S1E). Furthermore, immunostaining experiments confirmed this heterogeneity—after the 1st activation peak, the phospho-ERK signal displayed a higher average and amplitude spread for NGF in comparison with EGF (Fig EV1A–C). Comparison of population-averaged ERK activity measurements using Western blot (Fig 1D), or

immunofluorescence (Fig EV1B), versus EKAR2G (Fig 1E), revealed somewhat slower desensitization kinetics of the 1st ERK activity peak for the latter. Immunofluorescence analysis of native versus EKAR2G-expressing PC-12 cells revealed a slight lag of 1st peak phosphoERK desensitization kinetics in the latter cells (Fig EV1D), without however affecting the amplitudes of EGF versus NGF phosphoERK signals after the first peak. This indicates that biosensor expression affects the MAPK signaling network to some extent. Biosensor FRET ratio measurements do not necessarily scale linearly with the signaling events they report on, and it has previously proven valuable to explicitly model this (Birtwistle *et al*, 2011; Fujita *et al*, 2014). However, given the strong similarity between the Western blot, immunofluorescence, and EKAR2G datasets, we assumed that FRET ratio measurements could be used directly as a proxy for ERK activity. These results indicate that, while EKAR2G expression has a slight impact on a specific phase of ERK activation dynamics, it remains a faithful indicator of ERK signaling.

To quantify cell heterogeneity, we pooled all high- and low-dosage, EGF- and NGF-triggered ERK activation trajectories described above (Fig 1E–H), and used k-means clustering to extract 5 representative temporal activation patterns (Fig 2A). We then determined their incidence in response to the different GF dosages (Fig 2B and C) and found that EGF favored adaptive responses, while NGF led to a larger number of sustained ERK activity trajectories. We also observed ultrasensitive and switch-like ERK responses for increasing dosages of EGF and NGF. The 1st peak amplitudes saturated and remained unchanged above threshold concentrations of 1 ng/ml EGF and 2 ng/ml NGF (Fig EV2A and B). At lower concentrations, lower and more heterogeneous 1st ERK activity peak amplitudes were observed. A striking observation was that the different representative ERK activity trajectories displayed distinct 1st ERK activity peak amplitudes that correlated with the ability to produce a transient or a sustained response—cells with low or high 1st peak amplitude, respectively, produced transient or sustained responses (Fig 2A–D). For additional quantification of this phenomenon, we correlated 1st peak amplitude with long-term, 40′, ERK activity (indicating transient or sustained ERK activity). Robust correlation was observed for high and low NGF dosages, suggesting that high 1st ERK peak amplitude leads to sustained responses (Appendix Fig S2). Non-significant correlations were observed for high and low EGF dosages. Another interesting observation was that a low EGF dosage significantly shifted the distribution of ERK activity trajectories toward more sustained profiles when compared to the high EGF dosage (Fig 2A–C). Importantly, such subtle shifts in the distribution of ERK activity profiles are not apparent when a population average is computed (Fig 1G), illustrating the value of our clustering approach to analyze our single-cell trajectories datasets. Together, these results show that GF stimulation produces heterogeneous ERK activity responses, with both GFs being able to induce transient and sustained responses in different cells of the population. This most likely results from different strengths of EGF-triggered negative and NGF-triggered positive feedbacks within different cells. In the case of NGF, the presence or absence of sustained ERK activation responses might depend on the strength of the positive feedback (Ferrell & Machleder, 1998), which is consistent with the finding that the amplitude of the 1st peak correlates with the establishment of a sustained response. Sustained responses might, however, also result from lack of adaptation through negative feedback and continuous signaling input due to constant GF exposure.

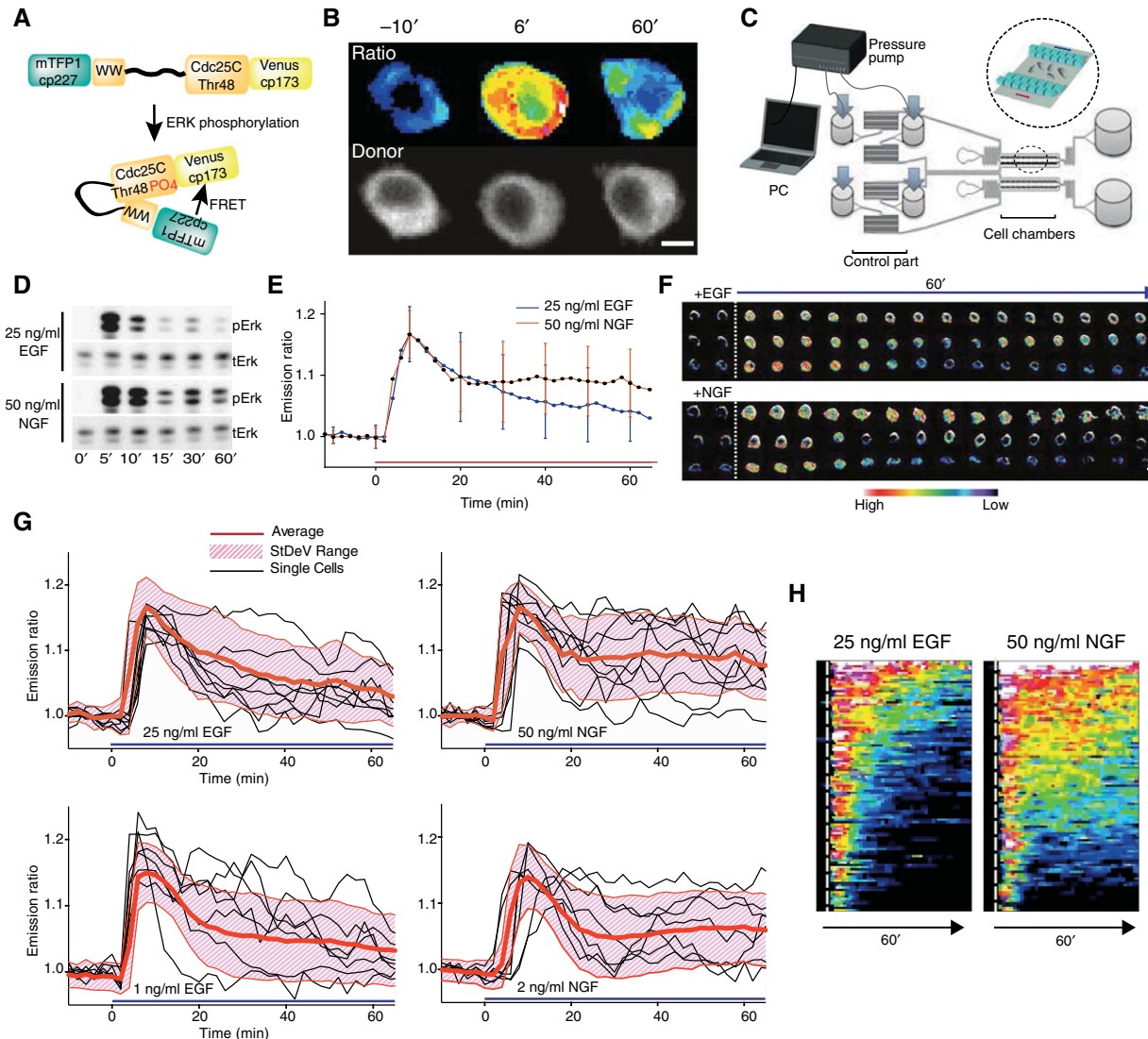

**Figure 1. Sustained GF stimulation induces heterogeneous ERK activity dynamics.**

A  EKAR2G biosensor. Upon phosphorylation by ERK, binding of the WW phospho-recognition domain to the phosphorylated Cdc25 substrate sequence leads to spatial re-orientation of donor (mTFP1) and acceptor (Venus) fluorophores leading to a FRET change that can be ratiometrically measured.

B  Ratiometric and mTFP1 donor images of EKAR2G in an EGF-stimulated PC-12 cell at the indicated time points, with $t = 0'$ corresponding to EGF application. Upper panel: FRET ratio image is color-coded for ERK activity. Lower panel: raw mTFP1 donor image in black/white contrast. Scale bar = 20 μm.

C  Flow-based, microfluidic device for temporal GF delivery. Computer-controlled, pressure pump enables mixing of medium and GFs in the control part (left), and temporally defined GF delivery in the cell culture chamber (right and magnified inset).

D  Population average of ERK activation dynamics measured by Western blot using a phosphoERK antibody.

E  Population average of ERK activation dynamics cell-averaged EKAR2G emission ratios (ERs) from $n=$ at least 111 cells. StDev are shown.

F  Selected EKAR2G ratio time series illustrating, from top to bottom, sustained, oscillatory, or transient ERK activity dynamics. ER is color-coded as in bar. Scale bar = 20 μm.

G  Single-cell ERK activity trajectories. Cell-averaged ERs for $n = 10$ cells, standard deviation range (StDev), population average for the indicated GFs and dosages. Experimental time courses were normalized to the mean of 5 time points immediately preceding GF application.

H  Waterfall plots of single-cell ERK activity trajectories. Cell-averaged ER trajectories are color-coded ($n = 78$ cells), population average (bottom). Vertical dotted line indicates GF application.

Source data are available online for this figure.

## Pulsed GF stimulation reveals novel features of the MAPK network

Because continuous signaling input does not effectively probe the MAPK network to produce adaptive or bistable responses

(Kholodenko *et al*, 2010; Avraham & Yarden, 2011), we applied single GF pulses using our microfluidic device. A high-dosage, 3′ and 10′ EGF, or 3′ NGF pulse elicited a robust peak of ERK activity that immediately adapted (Fig 3A and B, and Video EV2). In contrast, a 10′ NGF pulse induced sustained ERK activity within

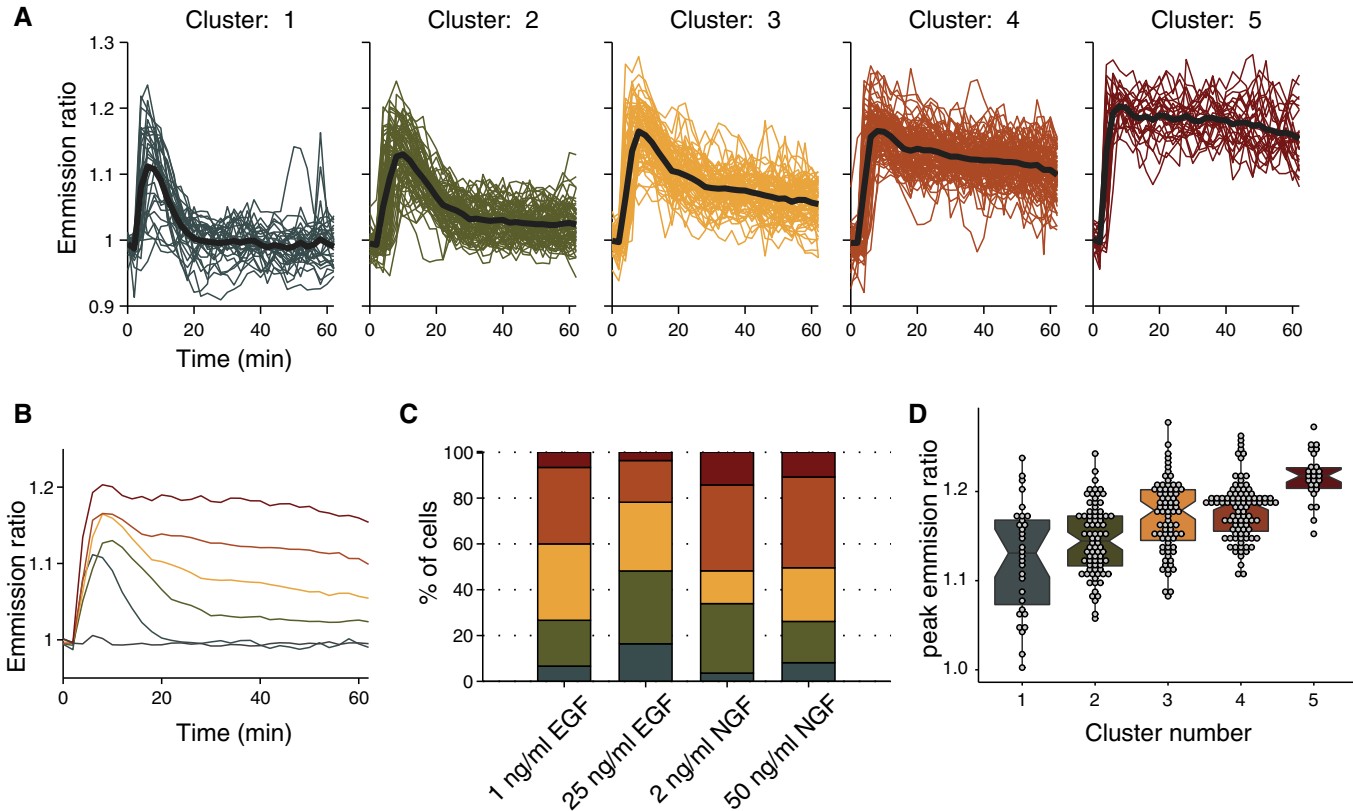

**Figure 2.  Quantification of ERK activity trajectory heterogeneity.**

A   Trajectories from all sustained GF stimulation experiments (*n* = 307 cells, same ERK activation trajectories as in Fig 1) were pooled and five representative trajectories were identified using k-means clustering with squared Euclidean distance. Raw (color-coded by cluster) and cluster representative trajectories (black).
B   Overlaid cluster representative trajectories.
C   Population distribution of representative ERK activity trajectories in response to different GF dosages. Data representative of *n* = 3 experiments.
D   Peak emission ratio intensity for each cluster. Boxplots with median, interquartile (box) and 1.5 IQR (whiskers) range, and raw datasets are shown. Boxplot notches extend 1.58 IQR / $\sqrt{N_{obs}}$, which gives approximately 95% confidence interval for comparing medians.

a cell subpopulation, potentially indicating bistability (Xiong & Ferrell, 2003), while the remaining cells exhibited transient responses (Fig 3B). As for sustained NGF stimulation, cell trajectory clustering again indicated a correlation between first peak amplitude, and the ability to produce a sustained response (Appendix Fig S3). Low-dosage, 3′ and 10′, EGF and NGF single pulses all led to adaptive responses (Fig 3C and D). All GF pulses, as well as sustained GF stimulation, evoked a robust 1st ERK activity peak, with similar amplitude distributions across the different dosages, indicating switch-like responses (Fig 3E and F). In marked contrast with sustained GF stimulation, population-homogeneous, rapid desensitization kinetics were observed for pulsed GF stimulation, except for cells and conditions in which bistable behavior was triggered by NGF (Fig 3E and G). These results show that EGF only induces adaptive, while NGF can produce both adaptive and bistable outputs depending on input strength/duration.

To explore the timescale at which adaptation occurs, we retriggered ERK activity by a delayed, 2nd 3′ EGF or NGF pulse, which only can induce adaptive ERK activity transients. Retriggering after a 3′ delay, when ERK activity still peaked, did not affect ERK activation kinetics (Fig 4A). Retriggering after a 10′ delay,

when ERK activity already displayed significant desensitization, led to a 2nd ERK activity peak of lower amplitude (Fig 4B). Retriggering after a 20′ delay, when adaptation had occurred, enabled a 2nd activity peak of similar amplitude as the 1st one (Fig 4C). For both EGF and NGF, the MAPK network structure therefore makes ERK refractory to re-activation until adaptation has occurred.

In addition to the feedback loops to upstream components that operate on timescales of minutes, ERK also induces expression of dual-specificity phosphatases (DUSPs) which negatively regulate ERK on a timescale of hours (Patterson *et al*, 2009). Consistently, evaluation of ERK activity trajectories for 5 h after sustained GF stimulation displayed a global long-term trend to ERK adaptation for both EGF and NGF at high and low dosages (Appendix Fig S4). The heterogeneity inherent to these ERK responses makes it difficult to discern any patterns of regulation operating at the hour timescale. We therefore reasoned that repeatable GF pulses could trigger multiple ERK activity transients homogeneously across the cell population, as the refractory periods effectively reduced noise. Such a pulsing input might therefore provide a robust readout for long-term, hour-scale feedback loops. We therefore evaluated 3′ GF multi-pulse regimes with varying pause duration and GF

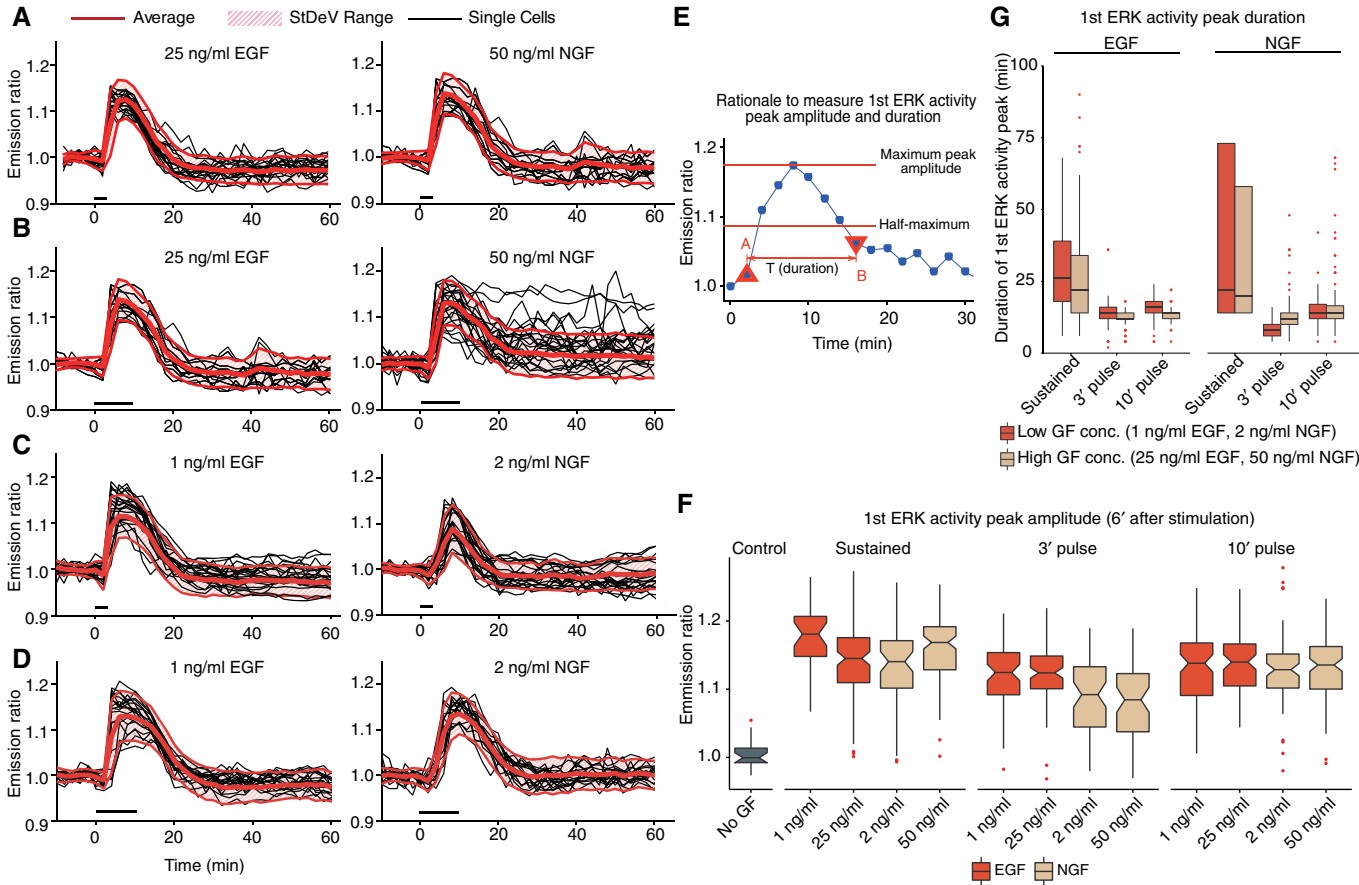

**Figure 3. ERK activity responses to single GF pulses.**

A–D  Single GF pulse experiments. Cell-averaged ERs, population average, and StDev range for *n* = 10 cells. Pulse application indicated by black bars. 3′ (A, C) and 10′ (B, D) pulse. High (A, B) and low (C, D) GF concentrations.

E  Rationale for measuring amplitude and duration of 1ˢᵗ ERK activity peak in response to GF stimulation. Peak amplitude was measured as the ER change from when the GF stimulation starts until the highest ERK activity before adaptation occurs. Peak duration was estimated as the time between the first point before reaching the half-maximum of the peak in the ascending phase, and the first point after the half-maximum in the descending phase.

F  Amplitude of 1ˢᵗ ERK activity peak in response to different sustained or pulsed GF stimulation experiments. Notched boxplots of 1ˢᵗ ERK activity peak amplitude with median, interquartile range (box), and data within 1.5 IQR range of the lower and upper quartiles (whiskers) are shown (*n* = 20 cells per experiment).

G  Duration of 1ˢᵗ ERK activity peak in response to sustained and pulsed GF stimulation. Boxplots of 1ˢᵗ ERK activity peak duration with median, interquartile range (box), and data within 1.5 IQR range of the lower and upper quartiles (whiskers) are shown (*n* = 20 cells per experiment).

Source data are available online for this figure.

dosages (Fig 4D–G, single-cell trajectories shown in Appendix Fig S5 and Video EV3). A high-dosage EGF, 3′ GF/3′ pause multi-pulse regime led to a robust single ERK activation peak that rapidly subsided, while the identical NGF pulse regime exhibited diminished ERK desensitization (Fig 4D). The 3′/10′ and 3′/20′ regimes triggered well-resolved ERK activity transients whose amplitude diminished over time, to a higher extent for EGF than for NGF (Fig 4E and F, and Video EV4). Lower desensitization appeared for the low versus the high EGF dosages, while this difference was much smaller for high and low NGF dosages (Fig 3D–F). The 3′/60′ regimes did not lead to pronounced long-term ERK activity desensitization (Fig 3G). Figure 3H quantifies the hour-scale decay of ERK activity maxima. These results clearly identify distinct hour-scale negative feedbacks that depend on GF concentration and identity and that are not distinguishable using sustained GF stimulation.

## Modeling MAPK network structure

Dynamic responses of the MAPK signaling to sustained GF stimulations have been extensively modeled in the literature (Kholodenko, 2000, 2006; Santos *et al*, 2007; von Kriegsheim *et al*, 2009; Kholodenko *et al*, 2010; Nakakuki *et al*, 2010). Our dynamic probing of ERK signaling using GF pulses in single living cells provides for a unique opportunity to calibrate the models to a much higher detail and study the dynamics at temporal scales inaccessible through standard population-averaged measurements. This allowed us to refine established models by incorporating novel feedback and crosstalk structures that are required to explain the salient features identified in our experiments (Fig 5A). Our core model consists of the classic Raf/MEK/ERK cascade with a short-term (on the time-scale of minutes) negative feedback from ERK to Raf that acts for both EGF and NGF stimulations, and a short-term positive feedback

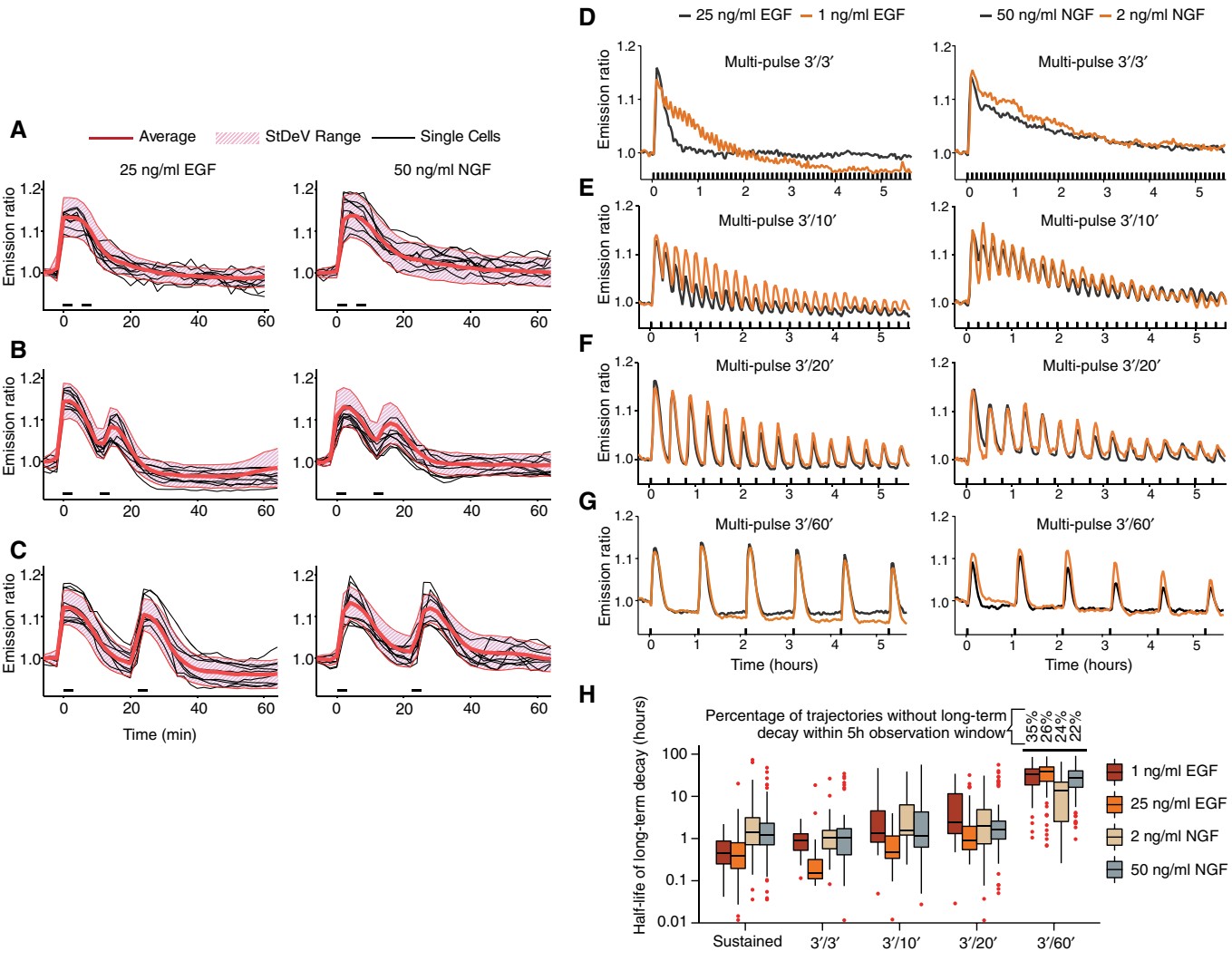

**Figure 4.  ERK activity responses to multiple GF pulses.**

A–C    Double-pulse GF experiments. (A) 3′ GF/3′ pause/3′ GF. (B) 3′ GF/10′ pause/3′ GF. (C) 3′ GF/20′ pause/3′ GF. Cell-averaged ERs, population average, and StDev range for *n* = 10 cells.

D–G    Multi-pulse GF experiments. Population average ERs (*n*= at least 30 cells) for different EGF/NGF dosages. (D) 3′ GF/3′pause. (E) 3′ GF/10′ pause. (F) 3′ GF/20′ pause. (G) 3′ GF/60′ pause.

H    Decay kinetics of ERK activity maxima from (D–G). Boxplots with median, interquartile (box), and 1.5 IQR (whiskers) range are shown for at least *n* = 30 cells.

Source data are available online for this figure.

from ERK to Raf only during NGF stimulation (Santos *et al*, 2007). An additional negative feedback via the expression of DUSPs that inactivate ERK through dephosphorylation operates on the timescale of hours (Nakakuki *et al*, 2010; Fey *et al*, 2012) (Fig 5A).

Two extensions of the core model are critical for explaining the salient features revealed by our pulsing experiments. First, in the 3′/3′, 3′/10′, and 3′/20′ multi-pulse EGF regimes, we observed that the high- and low-dosage EGF responses diverge for the later time points even though the 1[st] ERK activity peak amplitudes are identical (e.g., the 1[st] 20′) (Fig 4D–G). It is well established that ERK desensitization in response to EGF is caused by the activation of negative feedback loops (Santos *et al*, 2007; Fritsche-Guenther *et al*, 2011; Fey *et al*, 2012). However, the fact that the 1[st] ERK activity peaks are identical in response to the different EGF dosages implies

that negative feedback alone cannot account for the observed long-term differences—a pure feedback mechanism exclusively depending on ERK would be activated to the same amount. Thus, we explicitly introduced an additional feed-forward component, NFB, whose activation depends on both active ERK and receptor activity (Fig 5A). Second, we observed that a high-dosage 10′ NGF pulse elicited sustained ERK activity in a cell subpopulation, suggesting the presence of bistability (Fig 3B, Appendix Fig S3). In contrast, high- and low-dosage 3′ NGF and low-dosage 10′ NGF single pulses only elicited transient, adaptive ERK activity profiles throughout the cell population (Fig 3A–D). Considering that all pulsed NGF stimulation conditions led to an identical 1[st] ERK activity peak implies the existence of (i) a slight delay in the activation of the NGF-evoked positive feedback and (ii) an additional regulation of the ERK-Raf

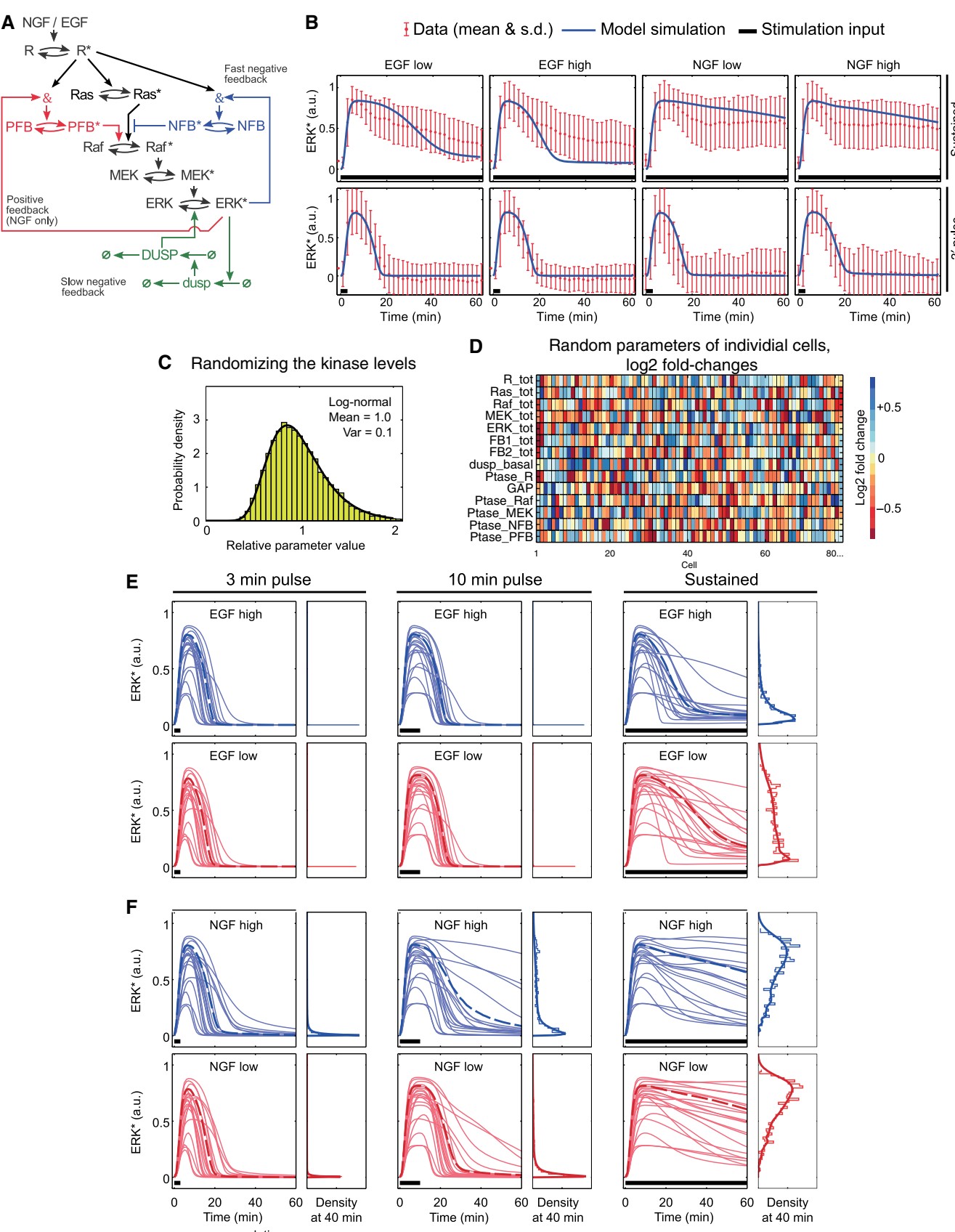

**Figure 5.**

**Figure 5. Deterministic and semi-deterministic modeling of sustained and pulsed GF stimulation.**

A Model topology scheme. Asterisks indicate active proteins. Blue/red colors indicate wiring in response to EGF/NGF stimulation. &: feedback loop integration, ø: synthesis and degradation, NFB/PFB: protein mediating negative/positive feedback.

B Deterministic modeling of sustained and 3′ pulse GF stimulation. Normalized, averaged ER trajectory from EKAR2G ERK activity measurements with means (dots) and StDev (error bars) are shown in red. Simulated ERK activity (ERK*) curves are shown in blue. Black line represents GF stimulation input. The model was simulated with a single parameter set (Appendix Table 1). Sustained and 3′ pulse GF stimulation ERK activity trajectories are from Fig 2 and Fig 3A and C, respectively.

C, D An ensemble of cells was modeled by randomly distributing the parameter values for all kinase, feedback protein, and phosphatase concentrations according to a lognormal distribution shown in (C). Each individual cell is characterized by slightly different parameter values as illustrated in (D), where each column corresponds to one cell.

E, F Ensemble modeling of ERK activation dynamics in response to 3′/10′ pulsed or sustained, high- and low-dosage GF stimulation. An ordinary differential equation (ODE) model was run 1,000 times with total signaling component concentrations drawn from the log-normal distribution shown in (C). (E) EGF, (F) NGF.

positive feedback strength, which would explain the induction of adaptive or sustained ERK activation. In the model, we accounted for the delay by including a positive feedback component, PFB, with kinetics on the observed timescales, and made the activation of this component dependent on receptor activity (Fig 5A). Note that the slight delay is caused implicitly by the activation of this feedback component, which takes time, not by introducing an explicit delay equation. A detailed account of all reactions and equations of the model is provided in Appendix Table S1.

**Benchmarking model simulations against experimental data**

To validate our model, we benchmarked simulated GF-evoked ERK activity outputs against experimental data. Using a purely deterministic approach with a fixed parameter set, our model captured the average behavior of the measured ERK activity trajectories in response to both sustained and 3′ pulse EGF and NGF stimulation (Fig 5B). However, the simulations fitted the experimental population average much better in case of a 3′ pulse than sustained GF stimulation. The reason being that sustained stimulation induces highly heterogeneous cell responses that deterministic modeling cannot account for. In contrast, cell responses due to a 3′ GF pulse are more homogeneous and hence better represented by the population average.

To take into account cell heterogeneity, we performed semi-deterministic modeling that was based on the same deterministic equations, but with individual cells simulated with total protein concentrations of signaling components randomly chosen from a log-normal distribution that corresponds to the distribution of protein abundances across the cell population (Fig 5C and D). Specifically, we sought to explain the emergence of sustained ERK activation in response to a 10′ high-dosage NGF pulse in a cell subpopulation, versus the population-homogeneous, transient responses observed in all other EGF/NGF pulsed stimulation conditions (Fig 3A–D). Simulating the ensemble model mimicked the experimental data in all the single EGF and NGF pulse conditions tested (Fig 5E and F). In the case of the 10′ NGF pulse, k-means clustering of simulated ERK activity trajectories indicated a correlation between initial ERK peak activity amplitude, and sustained ERK activity response (Appendix Fig S6), which recapitulated our experimental data (Appendix Fig S3). In contrast, low initial ERK peak activity amplitude led to adaptation. This suggests that differences in signaling component expression levels across the population might induce some cells to display high initial ERK peak activity leading to sufficiently strong positive feedback to induce bistable ERK activation (Ferrell & Machleder, 1998; Santos *et al*,

2007). However, the inability of the system to sustain bistability over time can be explained by the slow-acting, DUSP-catalyzed ERK dephosphorylation. Removing ERK-induced DUSP expression from the model enabled robust bistable ERK activation. Together, these results show that, depending on the heterogeneity in abundance of signaling component in different cells of the population, the network is capable of inducing adaptive and sustained ERK activity outputs in response to a transient NGF stimulus. We also modeled sustained GF stimulation using semi-deterministic modeling and found that our model captured several aspects of our experimental datasets (Fig 5E and F). The model captured the emergence of more sustained ERK activity trajectories in cells stimulated with low versus high EGF concentrations, which arises through modulation of negative feedback through receptor activity (compare distributions of simulated ERK activity trajectories in Fig 5E with experimental ones in Fig 2B). These results again show that heterogeneity in abundance of signaling components leads to emergence of distinct cell behaviors in the population.

To confirm that our proposed model structure did not contain unnecessary regulatory elements, we benchmarked simulations from alternative model structures against experimental data. A model in which the feed-forward crosstalk was removed could not explain the differences between the low and high NGF dosage, 10′ pulse stimulations (Fig EV3A and B). A model in which the delay introduced by the negative feedback component and the feed-forward crosstalk have been removed by making the Raf activation directly dependent on active ERK could not explain the differences between the 3′ and 10′ pulse stimulations (Fig EV3C and D).

We then used deterministic modeling to benchmark our model against the GF multi-pulse experimental datasets, in which homogeneous cell responses are faithfully reflected by the population average. Consistently, the deterministic model calibrated to population-averaged time courses replicated the multi-pulse experiments (Fig 6A). This captured the different hour-scale ERK activity amplitude decays observed experimentally in response to different GFs and dosages (Fig 4D–G), which can be explained by ERK-induced expression of DUSP phosphatases, producing a slow-acting negative feedback dephosphorylating ERK. To further test the relevance of the DUSP module, we removed it from the network and observed a loss of long-term adaptive behavior (Fig EV4A and B). The observation that 3′/3′, 3′/10′, and 3′/20′ EGF multi-pulse regimes caused the stronger decay of the ERK activity amplitudes at high versus low EGF dosages arises from the regulation of the negative feedback strength by the receptor activity. Consistently, a model in which this interaction is removed failed to explain the decrease of ERK activity peak amplitude observed in response to high versus low EGF dosage

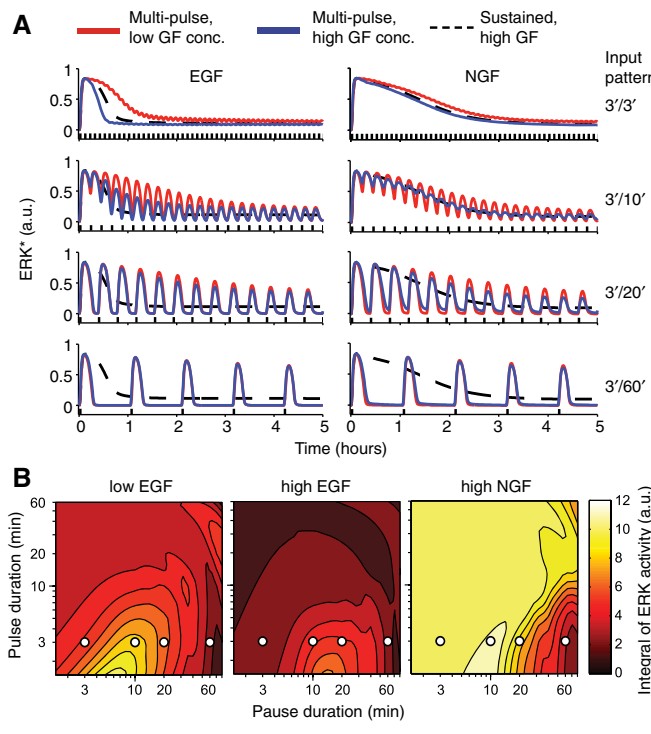

**Figure 6. Modeling of multi-pulse GF stimulation.**

A  Deterministic modeling of responses to multi-pulse EGF/NGF stimulation at different dosages.

B  Integral of ERK activity in response to different pulse/pause regimes calculated from the ODE model for a duration of 60′, 90′ after GF stimulation. White dots indicate conditions, for which cell differentiation is evaluated.

(Fig EV4C and D). In particular, the loss of explanatory power in this model could not be recovered by changing the values of the negative feedback parameter.

We also explored the potential involvement of EGF receptor internalization and degradation (Avraham & Yarden, 2011), to test whether this process could be involved in the long-term desensitization observed in high versus low EGF dosages in multi-pulse experiments. We therefore removed the receptor cross-talk to the negative feedback and replaced it by explicitly modeling receptor degradation and internalization using a range of degradation parameter rates (Fig EV5). We found that blocking internalization or changing internalization/degradation rates affected ERK activation dynamics in multi-pulse experiments, but in an EGF dosage-independent fashion. Thus, EGF receptor internalization/degradation cannot account for the ERK activity responses we observe experimentally.

Finally, because a receptor activity-dependent, feed-forward component that modulates the strength of the positive feedback (Fig 5A) was important to explain both the heterogeneous, adaptive, or bistable ERK activation responses in response to single NGF pulses (Fig 5C–F), as well as the NGF multi-pulse datasets (Fig 6A), we analyzed it more in detail. More specifically, we explored different modalities, linear dependency versus hill-shaped kinetics, for modeling the strength of this feed-forward component (Appendix Note 1). We observed that these different modeling

modalities did not affect the output of model-simulated ERK activity trajectories in response to both single 3′ and 10′ single high- and low-dosage NGF pulses, as well as our multi-pulse datasets (Appendix Fig S7). Together, our combined experimental/modeling approach strongly supports the proposed structure of our model.

**Rewiring cell fate using pulsed GF stimulation**

The integrated ERK activity over time, ERK$^{\text{INT}}$, has been correlated with cell fate determination in different cell systems (Murphy *et al*, 2002; Santos *et al*, 2007; Albeck *et al*, 2013; Aoki *et al*, 2013). We reasoned that synthetic GF multi-pulse regimes might precisely manipulate duration of ERK signaling, and ultimately the differentiation fate. Using our model, we predicted temporal GF pulse/pause regimes that optimize ERK$^{\text{INT}}$ levels (Fig 6B), and found that distinct GF pulse regimes lead to different ERK$^{\text{INT}}$ levels. Long-term EKAR2G imaging induced phototoxicity that hampered differentiation, precluding direct ERK$^{\text{INT}}$/differentiation correlation at a single-cell level. We therefore subjected cells to different pulse regimes for 13 h followed by additional 24 h in GF-free medium, and quantified steady-state neurite outgrowth as a measure of differentiation (Fig 7A and B) using an automated neurite/soma segmentation algorithm (Appendix Fig S8A). Statistical analysis of this dataset is also shown (Appendix Fig S8B). As expected, sustained EGF and NGF stimulation, respectively, led to undifferentiated and differentiated states. 3′/3′, 3′/10′, 3′/20′ high NGF dosage multi-pulse regimes led to a statistically significant increase in differentiation compared to sustained NGF stimulation, while the 3′/60′ regime did not lead to differentiation at all. While not as robust as NGF in terms of neurite length (and thus differentiation ratio), we found that 3′/10′ and 3′/20′, but not 3′/3′ nor 3′/60′, high and low EGF dosage multi-pulse regimes induced a potent differentiated state. However, neither sustained application of low NGF nor a 3′/10′ low NGF multi-pulse regime led to robust differentiation, despite the fact that these stimulation protocols lead to similar ERK$^{\text{INT}}$ levels than with high NGF dosage. This suggests that high NGF dosage is required to trigger additional signaling pathways that are important for neuronal differentiation and/or survival. Our experiments show that, except for low NGF concentrations, model-predicted ERK$^{\text{INT}}$ optimization using synthetic EGF or NGF multi-pulse regimes with specific frequencies lead to robust differentiation. In the case of EGF, such frequency-modulated synthetic multi-pulse regimes therefore convert a proliferation into a differentiation signal, and allow for control of cell fate independently of GF identity. Maintenance of sustained ERK activity through synthetic multi-pulse regimes most likely enables for IEG product stabilization through their phosphorylation that can then instruct the differentiation fate (Murphy *et al*, 2002; Nakakuki *et al*, 2010). The loss of any differentiation in response to stimulation with 60′-spaced adaptive ERK activity transients can be explained by the very short-lived nature of IEG mRNAs and proteins (Murphy *et al*, 2002; Avraham & Yarden, 2011).

# Discussion

We show that the classically studied EGF-triggered transient and NGF-triggered sustained ERK activation dynamics are highly

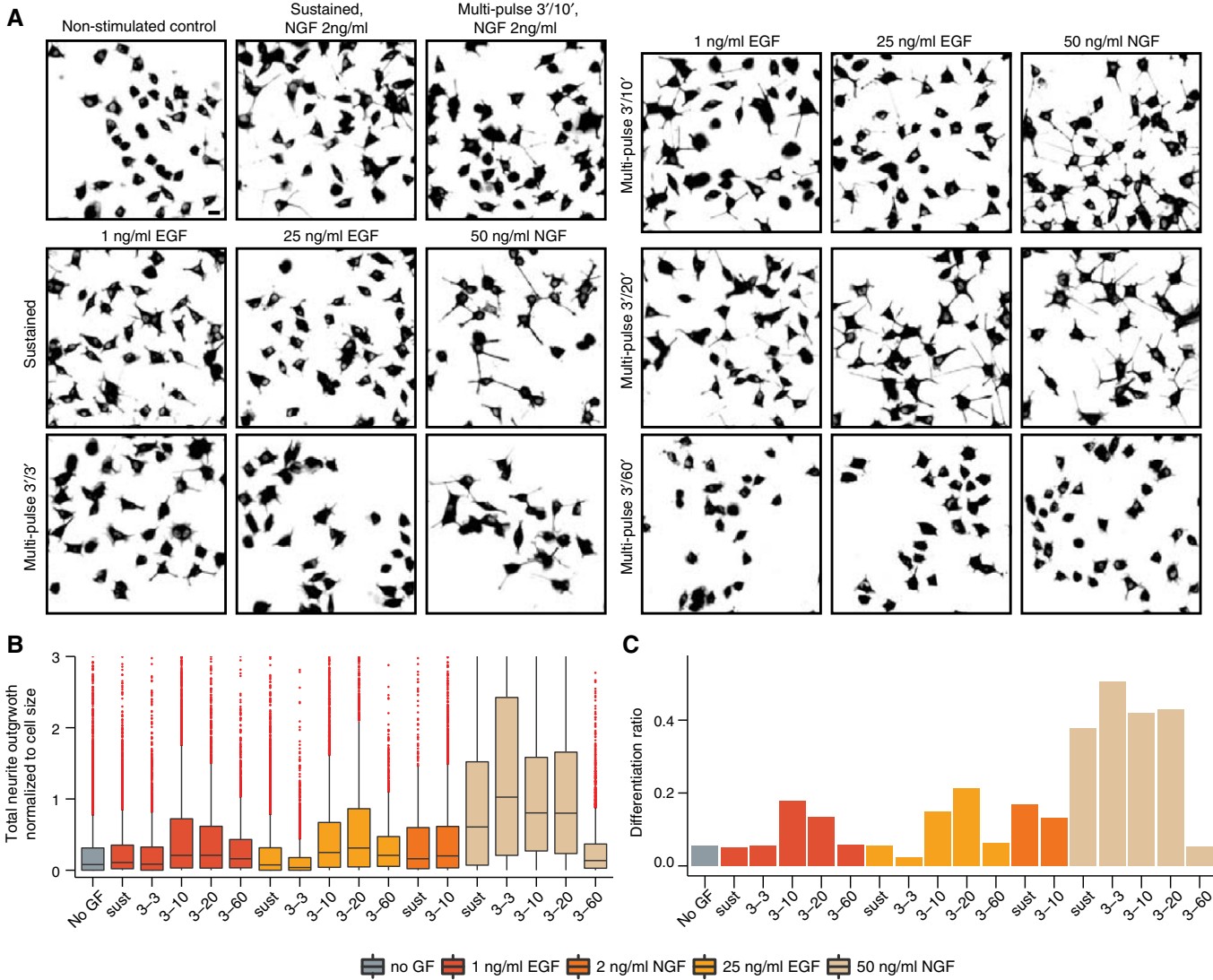

**Figure 7.    Manipulation of ERK activation dynamics rewires cell fate.**

A    Representative micrographs of tubulin-stained PC-12 NS-1 subclone cells stimulated with sustained and multi-pulse GF regimes. Scale bar = 20 μm.

B, C    Quantification of neuronal differentiation fate. (B) Total neurite outgrowth per cell normalized to the major axis length of cell soma for sustained and multi-pulse GF regimes. (C) Differentiation ratio calculated as the fraction of cells with normalized total neurite outgrowth per cell larger than 1. At least 5,000 cells quantified in at least 2 independent experiments for each condition. Boxplots with median, interquartile (Box), and 1.5 IQR (Whiskers) are shown.

heterogeneous when single cells within a population are analyzed. Our single-cell measurement approach therefore clearly enhances previous models based on cell population averages (Marshall, 1995; Santos et al, 2007; von Kriegsheim et al, 2009; Shin et al, 2009; Nakakuki et al, 2010). Important problems encountered with sustained growth factor application are that heterogeneous ERK activity dynamics responses preclude the use of deterministic modeling, and do not allow us to properly evaluate MAPK network properties such as adaptation and bistability. Dynamically probing the MAPK signaling flux by applying pulsed growth factor inputs solves both issues. This allowed us to uncover novel features of the MAPK network topology that consists of different feedback and feed-forward loops acting at different timescales.

Our study clearly indicates the existence of feed-forward networks by which receptor activity regulates the strength of the positive and negative feedback loops to fine-tune ERK activity. This is consistent with a recent report in which EGF-triggered ERK dynamics regulating cell cycle entry required receptor-dependent signaling events in addition to the feedback loops operating within the core Ras-Raf-MEK-ERK cascade (Sparta et al, 2015). The most likely candidate feed-forward mechanisms that impinges on the positive feedback loop from ERK to Raf involves protein kinase C (Santos et al, 2007) and PLCγ (von Kriegsheim et al, 2009). For the negative feedback loop, candidates include the phosphorylation of Raf at inhibitory sites and the action of as of yet unidentified kinases (reviewed in (Matallanas et al, 2011)), or the AKT-mediated disruption of the PEA-15–ERK complex, which is part of the NGF

differentiation program (von Kriegsheim *et al*, 2009). Concerning the latter, release from the PEA-15 complex would relieve competitive inhibition and allow ERK to phosphorylate several upstream negative feedback sites. Furthermore, recent proteomic analysis of the dynamic, EGF-triggered, interactome of the adaptor Shc1 (Zheng *et al*, 2013) has identified a temporally regulated AKT feedback phosphorylation of Shc that recruits the tyrosine phosphatase Ptpn12, ultimately leading to EGF receptor dephosphorylation, Grb2 dissociation, and signal termination. All these possibilities provide attractive means to translate strength of receptor activity into modulation of negative and positive feedback, and will have to be systematically studied in the future.

One important insight enabled by our study is that EGF solely leads to adaptive ERK activation (in the absence of continuous signaling input), while NGF can generate both adaptive and bistable outputs depending on growth factor input strength/duration and on population signaling heterogeneity. The latter feature possibly arises from stochastic expression levels of signaling components as shown by our modeling studies. As previously proposed (Chen *et al*, 2012), this might enable the MAPK network to maintain a homeostatic balance of cell fates in response to NGF, preventing excessive proliferation or differentiation, which would be detrimental to multicellular organisms.

We also find that re-triggering the MAPK network multiple times, while maintaining it in an adaptive regime, homogenizes the signaling state across the cell population. This proved essential to uncover feedback signaling mechanisms that occur on hourly timescales that are difficult to discern in heterogeneous cell responses in response to sustained growth factor stimulation. Based on our enhanced model of the MAPK signaling network, optimization of integrated ERK activity by delivery of GF pulse regimes at specific frequencies can then be used to efficiently rewire cell fate decisions independently of GF identity, in the absence of any drugs or gene perturbations. Thus, stimulation with EGF pulses delivered at the required frequency to optimize integrated ERK activity leads to differentiation rather than proliferation. This might have a number of biotechnological applications. However, we also found that in some instances, GF pulsing schemes that lead to identical integrated ERK activity do not necessarily lead to the same level of differentiation (Fig 7, high versus low NGF dosages). Not surprisingly, this suggests that depending on the GF, additional signaling networks than ERK are important for cell fate decisions.

We foresee that our pulsed growth factor stimulation strategy, which allows us to bypass cellular heterogeneity, probe for specific network properties, and identify feedback loops that operate at different timescales, will be a powerful tool to identify the precise molecular circuitry in the MAPK network, when combined with perturbation approaches.

# Materials and Methods

### Cell culture and generation of stable cell lines

PC-12 (American tissue culture collection) and its Neuroscreen-1 subclone (a kind gift from Tobias Meyer) cells were cultured in DMEM (Sigma, Germany) supplemented with 10% horse serum, 5% fetal bovine serum, and 1% penicillin/streptomycin. Cells were tested for mycoplasma contamination using polymerase chain reaction-based kits. Cells were cultured on 50 μg/ml collagen solution from bovine skin (Sigma, Germany). For that purpose, tissue culture dishes (BD falcon, USA) were collagen-coated at room temperature for 30 min before cell seeding. At 70% confluence, cells were gently detached using a Cell scraper and passaged. Due to their decreased propensity to aggregate, Neuroscreen-1 cells were used in experiments in which cell fate was analyzed. This greatly facilitated neurite outgrowth analysis.

For stable cell line generation, we produced lentiviral particles expressing the ERK activity biosensor EKAR2G1 (Fritz *et al*, 2013). Briefly, HEK293 FT cells (Invitrogen) were transfected with lentiviral and packaging constructs. Supernatant was then collected and concentrated using a Lenti-X Concentrator Kit (Clontech). PC-12 cells were infected and selected with 0.5 μg/ml puromycin until a stable cell population appeared. Cells were subsequently cultured in the presence of 0.5 μg/ml puromycin, which was removed before imaging experiments. PC-12/EKAR2G cells were used for imaging ERK activation dynamics.

### Construction and handling of microfluidic devices

We used a modified version of a previously described microfluidic circuit to apply chemokine gradient to cells (Lee *et al*, 2012). Slight modifications of the construction of the microfluidic circuit allowed us to deliver GF pulses. The microfluidic silicon master was replicated from a silicon wafer with SU-8 microstructures. The silicon master mold was composed of two 40-μm- and 100-μm-thick layers of photoresist. First, the plasma-treated silicon wafer was spin-coated with SU-8 100 (Microchem, USA) negative thick photoresist until a height of 40 μm was achieved. After baking at 65°C for 5′ and 95°C for 20′, the wafer was exposed to 405-nm ultraviolet light (Shinu MST, Korea) with a 250-mJ dose and masked by the negative film mask (Han&All Tech, Korea). After this 1st round of exposure, the wafer was baked again at 65°C for 1′ and 95°C for 10′. SU-8 developer (Microchem, USA) was then used to remove the unexposed part of the photoresist. For deposition of the 2nd photoresist layer, the film mask for the second master was correctly positioned using the alignment pattern on the first layer of the wafer. The photoresist for the second layer was then spin-coated until a 100-μm thickness was achieved. The wafer was baked at 65°C for 10′ and 95°C for 30′ and exposed to 500 mJ of 405-nm UV light. After a final baking step at 65°C for 1′ and 95°C for 10′, the wafer was dipped into the developer, and baked to evaporate the residual solvents on the top.

Poly-dimethylsilosane (PDMS) was used to replicate the master. The precursor (Sylgard 184, Dow Corning) was mixed at a 10:1 ratio and degassed in a vacuum chamber for 5 min. About 7 g precursor was then poured on the top of the master, and solidified at 80°C in a dry oven for 30′. The plastic reservoir from a 8-well strip (Evergreen sci, USA) was then glued on top of the microfluidic device using precursor. These reservoirs contain the medium and growth factors and allow us to connect the microfluidic device to the ONIX pressure pump (Millipore). An additional layer of 30 g of precursor was added to seal the plastic reservoir. As shown in Fig 1C, the PDMS replica was cut and punched. Plasma treatment was used to bond the PDMS replica to the coverslip (Tasumi, Japan) to allow proper sealing that resists the high-pressure applied during the

experiments. To enhance the bonding strength, the device was heated for 5 min in a 80°C dry oven. After bonding, the device was immediately filled with PBS.

The microfluidic devices were coated using 50 μg/ml of collagen overnight at 4°C. Before cell seeding, devices were rinsed with PBS. PC12/EKAR2G or NS-1 cell suspensions were prepared at a concentration of $2 \times 10^6$ cells/ml. 50 μl of this cell suspension was added in the outlet and aspirated with a pipette from the cell reservoir inlet port (Fig 1C). After a 30′ incubation, residual cells in the outlet were removed by aspiration and the medium was replaced. Prior experiments, the cells were starved in DMEM containing 1,000 mg/ml glucose, 1% horse serum, and penicillin/streptomycin. Note that the flow is constant throughout the live cell experiments. This implies that after shear stress adaptation, GF-induced signaling responses can be reliably measured. Furthermore, there is a limit to the minimal GF pulse time that can be applied. Pulses that stably last less than 60 s require an amount of flow that will lead to cell detachment.

### Live cell imaging

All FRET ratio-imaging experiments were performed on an epifluorescence Eclipse Ti inverted fluorescence microscope (Nikon) with a Plan Apo air 20× (NA 0.75) objective controlled by Metamorph software (Molecular Devices). Laser-based autofocus was used throughout the experiments. Image acquisition was performed with a Hamamatsu Orca R2 camera at a 16-bit depth. Donor, FRET, and red channel images (to visualize an Alexa546-dextran that indicates GF exposure) were acquired sequentially using filter wheels. The following excitation, dichroic mirrors, and emission filters (Chroma) were used: donor channel: 430/24×, Q465LP, 480/40 m; FRET channel: 430/24×, Q465LP, 535/30 m; and mCherry channel: ET572/35, 89006bs, 632/60 m (for dextran imaging). Standard exposition settings were used throughout the experiments. 440-nm (donor and FRET channel excitation) and 565-nm (red dextran) LED lamps were used as light sources (Lumencor Spectra X light engine), with 1.1% (440 nm) and 1.5% (565 nm) of lamp power. Acquisition times were 300 ms for donor channel and 300 ms for FRET at binning 2 × 2. Cells were imaged in DMEM with 1,000 mg/ml glucose, 1% horse serum, and penicillin/streptomycin, at 37°C. The microfluidic device was mounted on the microscope stage and was connected by the tubing to the ONIX pump.

### FRET ratio image analysis

FRET ratio images produced with the EKAR2G biosensor were analyzed using a combination of ImageJ and Metamorph. Donor and FRET images were background-subtracted image by image. The FRET was divided by the donor channel, and multiplied by 1,000 to produce a 16-bit ratio image using the imageJ "calculator plus" plugin. The stacked ratio images were then transferred to Metamorph. Each cell was then segmented based on time averaged stack that provides a "footprint mask" for cellular movement of each cell, and stacks for each individual cells were created. Cell aggregates were manually segmented and discarded. The average ratio of each cell through time was measured using ImageJ, and saved in text files. Since every experiment involved a 2-h initial period to stabilize

cells to shear stress, we set the average intensity of 5 time points around 80′ after the flow has been switched on as the basal level ERK activation for normalization. The Alexa546-dextran channel was stacked and profiled through time to verify the accuracy of temporally defined growth factor exposure in the cell chamber of microfluidic device.

### NS-1 differentiation experiments

Prior to the differentiation experiments in microfluidic devices, cells were starved for 6 h. About 200 μl of GF-containing medium was then added to one of the input reservoirs (Fig 1C). To prevent evaporation, the cell inlet port was sealed using adhesive plastic. The microfluidic devices were connected through a syringe connector to an ONIX pressure pump (Millipore, USA). Differentiation experiments were performed in a table-top small incubator with connection ports (Galaxy, Eppendorf). The output valve pressure was stabilized at 1.5 psi and different constant or pulsatile protocols were run. Each protocol included a 2-h adaptation time to flow in starvation medium, followed by 13 h of sustained or pulsatile GF exposure, which was determined by changing the opening sequence of the computer-controlled pressure valve. Since there is a limitation in number of valves that could be controlled independently, we connected a 4-way conduit to each valve to maximize throughput. This strategy allowed us to use 8 valves embedded in the pump to process in 4 different microfluidic devices, ultimately allowing to perform 16 experiments simultaneously. Each chamber from a single microfluidic provided independent results for a given stimulation pattern. For each experiment, a negative (no GF) and a positive control (sustained stimulation with a GF) were included. After the 13-h GF stimulation period, devices were disconnected and incubated for an additional 24 h in the incubator in starvation medium.

Cells were then fixed with PBS, 4% paraformaldehyde (fixing solution). To access cells in the microfluidic device, 50 μl of fixing solution was dropped on the top of the cell inlet port, inducing flow through the cell chamber. After 15′, cells were permeabilized using PBS, 0.1% Triton X-100 for 30′ and blocked using PBS, 2% BSA in PBS for 15′. Cells were then incubated with an anti-alpha-tubulin (Sigma T9026, 1:1,000) antibody for 6 h, washed for 30′, and incubated with an Alexa-546 anti-mouse secondary antibody (1:1,000) and DAPI for 3 h. Samples were washed with PBS for 30′. DAPI and Alexa-546 images were scanned and stitched for the whole surface of each microfluidic lane using the Metamorph "scanslide acquisition" module. Identical immunofluorescence and imaging protocols were used to perform phosphoERK (1:1,000, Sigma M8159) and ERK2 (1:1,000, Sigma M7927) experiments. Secondary detection was then performed using Alexa-488- and Alexa-561-labeled antibodies.

Tubulin/DAPI-stained images from the differentiation experiments were processed using the Metamorph "Neurite Outgrowth Analysis" module (Universal Imaging). Fields of view in which a too high cell density was observed were manually erased using a mask. This allowed us to extract a number of metrics for each cell, such as total and average length of neurite outgrowth, and the number of neurites per cell. A segmented image with each cell containing an identification tag was then exported to the CellProfiler image analysis software (http://www.cellprofiler.org/) (Kamentsky et al, 2011). This was used to compute the major axis length of an

ellipse fitted to the cell soma. Ultimately, this was used to compute the fraction of differentiated cells with the criterion that a cell was considered differentiated, when the total neurite outgrowth was larger than the major axis length of the soma. It is important to mention that our computer vision-based neurite outgrowth segmentation approach was much more stringent than classic "manual" evaluation using the human eye (von Kriegsheim *et al*, 2009). A direct comparison of the quantification of the fraction of differentiated cells between these different methods is therefore not possible.

**Modeling**

Based on the scheme in Fig 5A, a dynamic model in terms of ordinary differential equations was derived. This model is consistent with established models in the literature (Kholodenko, 2000; von Kriegsheim *et al*, 2009; Nakakuki *et al*, 2010), which were used to compile parameter values used in this study. All phosphorylation and de-phosphorylation parameters were chosen to fit our experimental data, while adhering to biologically reasonable parameter bounds. The half-life of DUSP protein and mRNA was set to 90 min, which reflects the observed long-term decay in the measured ERK activity trajectories and lies within the experimentally reported range from 7.5 min to 4 h (Nakakuki *et al*, 2010; Bermudez *et al*, 2011; Cagnol & Rivard, 2013). EGF receptor internalization, recycling, and degradation were modeled as a two-step process using experimentally reported values for the internalization, degradation, and recycling parameters (Huang *et al*, 2006; Sorkin & Duex, 2010). The model was implemented and simulated by numerical integration in MATLAB. A detailed account of all reactions, equations, and parameters is provided in Appendix Table S1.

**Simulating possible alternative model structures**

Because the dynamic behavior of a model depends on its parameter values, adding or removing modules usually requires re-adjusting some parameter values in order to fit the experimental data (Conzelmann *et al*, 2004; Sauro & Kholodenko, 2004; Del Vecchio *et al*, 2008; Saez-Rodriguez *et al*, 2008). Therefore, when adding or removing a module, we analyzed a whole range of parameters (Fig EV5). The process itself is best explained using an example: Removing the feed-forward crosstalk from the negative feedback, while keeping all parameters fixed means that the strength of the feedback has changed. There is now a strong activation of the negative feedback, resulting in only one ERK pulse in response to multi-pulse EGF stimulation (Fig EV5B, panels on the right). Quite clearly, this does not match our experimental data and we have to allow for readjustments of the negative feedback parameters. Readjusting k3 from 0.0286/min to 0.0045 min recovered the observed response dynamics for low-dosage EGF stimulation (Fig EV5B, panels in the center), and readjusting to 0.001 min recovered the observed trajectories for EGF high (Fig EV5B, panels on the left). However, no single parameter readjustment was able to explain low and high EGF dosage conditions simultaneously. We can conclude that the receptor crosstalk is required to explain all of our data."

**Expanded View** for this article is available online.

## Acknowledgements

We are grateful to Tobias Meyer for providing the NS-1 cell line. This work was supported by a Human Frontier Program grant to O.P. and N.L.J, by the Brain Korea 21 Plus Project to N.L.J., by UCD Seed Funding and Science Foundation Ireland Grant No. 06/CE/B1129 to M.D. and B.N.K, by the EU FP7 PRIMES grant to B.N.K, by the Strategic Korean-Swiss Cooperative Program (Project reference: 2009-00525) grant to M.P. and N.L.J., and by a grant from the European Research Council to M.P.

## Author contributions

OP, HR, and MC designed experiments. HR, MC, and YB performed and analyzed experiments. MD, DF, and BNK performed data analysis and mathematical modeling. SSL and MP provided initial design of microfluidic device. HR, MC, and NLJ designed the second-generation microfluidic device. OP, MD, and DF wrote the paper.

## Conflict of interest

The authors declare that they have no conflict of interest.

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
