## [Review Process File · Molecular Systems Biology]

Frequency modulation of ERK activation dynamics rewires cell fate

Hyunryul Ryu, Mr. Minhwan Chung, Maciej Dobrzynski, Dirk Fey, Yannick Blum, Sung Sik Lee, Matthias Peter, Boris N Kholodenko, Noo Li Jeon, Olivier Pertz

Corresponding author: Olivier Pertz, University of Basel

Review timeline:	Submission date:	22 July 2015
	Editorial Decision:	19 August 2015
	Revision received:	19 September 2015
	Editorial Decision:	15 October 2015
	Revision received:	16 October 2015
	Accepted:	20 October 2015

Editor: Maria Polychronidou

Transaction Report:

1st Editorial Decision

19 August 2015

Thank you again for submitting your work to Molecular Systems Biology. We have now heard back from the three referees who agreed to evaluate your manuscript. As you will see from the reports below, the referees think that the presented findings are interesting and they appreciate the overall quality of the work. However, they raise a series of concerns, which should be carefully addressed in a revision of the manuscript.

The referees' recommendations are rather clear so there is no need to repeat all the points listed below. As you will see, reviewer #1 recommends extending the study by testing a few model predictions. While these additional analyses would indeed enhance the overall impact of the study and we would welcome their inclusion (i.e. if already available) we do not think that they are mandatory for the acceptance of the manuscript.

If you feel you can satisfactorily deal with these points and those listed by the referees, you may wish to submit a revised version of your manuscript. Please attach a covering letter giving details of the way in which you have handled each of the points raised by the referees. A revised manuscript will be once again subject to review and you probably understand that we can give you no guarantee at this stage that the eventual outcome will be favorable.

REFeree REPORTS

Reviewer #1:

The authors describe a study analysing MAPK signalling by using a microfluidics set up and life-cell single-cell readout of MAPK activity, and by challenging the network using a defined, pulse-

like stimulation pattern they can dissect the timescales shaping the dynamics of the network. The whole study is motivated by the paradigmatic differential response in PC12-cell differentiation, where stimulation with the growth factors NGF and EGF yield distinct responses of MAPK activity (prolonged vs. short) and lead to different phenotypes (neuronal outgrowth vs. proliferation).

The authors observe different short term response of EGF and NGF (which have been studied extensively), but more interestingly, they show that long-term adaptation (or desensitization) of the pathway depends strongly on the stimulus pattern. While sustained stimuli and rapidly repeated short stimuli result in lower amplitudes over time, stimuli intermitted by 60 min result in no decay for EGF stimulation, and reduced decay by NGF stimulation. By using a mathematical model they to build a plausible explanation for this effect: MAPK activity is mediated by fast positive (for NGF) and negative (for EGF) feedbacks, which are integrated in a feed-forward loop (converging at RAF). These two feedbacks together with the feed-forward loop generate the transient vs. sustained MAPK activity that distinguishes the different growth factors. Furthermore, a slow negative feedback via the induction of a phosphatase of the DUSP family causes the adaptive behaviour of the pathway on the time-scale of one hour. Finally, they show that with repeated 3min pulses of EGF (with braks larger then 10 min) they can trigger neurite outgrowth as the ERK kinetics is similar to NGF stimulation.

Overall, I think this is an excellent paper, is well written and such work of challenging the pathway with complex input pattern is timely and had to be done as it unveils important characteristics of the pathway's timescales and feedbacks. While I am no expert in life-cell imaging, I think the work has been well done and the data seems solid. The model seems to be well justified.

While the model plausibly explains the data, it was not really challenged and used to make predictions that could and should be tested experimentally. The prediction that neurite outgrowth can be triggered by pulsed EGF due to less adaptation/dampening at later time points is not really a model prediction but can be seen from the data. Therefore, I think having testable predictions from the model would strengthen the paper further. One of the main or instance, the main desensitizer in the model is the transcriptional induction of DUSP. As DUSP activity acts at the level of ERK, measuring (even only at a single or few time points for example by IF) ERK and MEK could be valuable to distinguish DUSP feedback and other (possibly slow) post-translational feedbacks. Also, adding a translation inhibitor may be a challenge to dissect the involvement of the slow DUSP feedback, although translation inhibitors may have more unspecific effects.

Minor: "NGF" is missing after 50 ng/ml in the legend of last figure .

Reviewer #2:

Ryu et al. use a combination of FRET reporter, microfluidics, and computational modeling to investigate the impact of ERK/MAPK signaling heterogeneity in PC-12 cells. This question is an important one, as the PC-12 system is a widely used example of how differences in the kinetics of ERK activity can result in different cell fates, but has still not been the subject of careful live-cell study of ERK kinetics. The authors describe substantial heterogeneity in response to both NGF and EGF, then go on to demonstrate that this heterogeneity can be attenuated by pulsed growth factor stimulation. After a thorough characterization of the pulse response kinetics for both growth factors, they construct a mathematical model, using this to understand the potential mechanisms underlying the different kinetics and to predict stimulation patterns that alter the differentiation response to EGF. These predicted patterns are then tested and shown to increase the differentiation response to EGF. Overall, this study addresses an important open question in the field, is elegantly and rigorously executed, and is beautifully presented. I found this study to be of great interest, and believe it will be a very valuable contribution.

There are a few issues that can be addressed by text or figure changes:

1) The interpretation of Fig. S3a as showing an "all or none" threshold between 0.5 and 1 ng/ml EGF or 1 and 2ng/ml NGF is a bit of a stretch. Certainly the dosage effect appears to be non-linear, but there is clearly some stimulation at 0.5 EGF or 1 NGF visible in the averages. The description of these data should be more in line with what is shown in the figure. Showing some single cell

trajectories (or a heatmap) here would be illuminating as to whether individual cells are showing a moderate response at lower EGF concentrations, or whether a few cells are responding with full pulses.

2) There is very little discussion of how the reporter, a substrate of ERK, relates to active ERK itself. In the model, these two species are directly equated. However, it is not necessarily the case that these quantities are linearly related (PMID 22114702). Other studies have found it necessary to explicitly model the reporter kinetics (PMID 24848561). It would be useful to justify the details of how this relationship is treated in fitting the model to the data.

Minor issues

Fig. 1b is referenced out of order in the text.

"NGF" missing from the legend in Fig. 6

Reviewer #3:

This single cell analysis of Erk response to EGF or NGF application regimes is an interesting work that nicely builds on previous knowledge on how distinct temporal Erk profiles that determine cell fate come about. For me the most interesting part is how receptor state affects via feed forward mechanism the positive feedback and negative feedback in the system. This makes the system dependent on the time of growth factor application, which allows the authors the manipulation of cell fate decisions by using pulsed growth factor application regimes. The feed forward mechanism that affects the positive feedback upon NGF stimulation has been previously described in Santos et al. (2007) as being dependent on PKC activity (To affect RKIP) and should be discussed in the text. The molecular mechanism of the feed forward that affects the negative feedback is new (and exciting) and is not discussed in terms of molecular mechanism. One point in relation to this feedback is that stimulation with a low dose of EGF should cause a sustained response because the negative feedback is not activated. Why does this only work in the pulsed EGF application regime? The authors should show the Erk response profiles for the sustained low EGF and NGF dose in Fig.1.

The other slight concern is that the results heavily rely on a substrate sensor of Erk activity (EKAR2G) whose distribution in the cell as well as its expression level might affect the interaction with phosphatases resulting in different inactivation kinetics. In other words: Erk activity is convoluted with the kinetics of EKARG2 phosphorylation/dephosphorylation that depend on its expression level and partitioning in the cell. The heterogeneity for example in the NGF response could arise from different dephosphorylation kinetics of EKAR2G in the nucleus versus cytoplasm. Can the authors distinguish the responses in these two cell compartments? Also, the fluorescence distribution of the sensor should be provided with the ratiometric images to judge its distribution in the cytoplasm versus nucleus. Even if the sensor rapidly exchanges, EKAR2G in the nuclear volume will dominate the signal. The way the ratiometric images look, one gets the impression that EKARG2 is actually excluded from the nucleus (I would also suggest to make Fig.1c larger to better see the responses). Is there a correlation between EKARG2 expression level and response?

The model (Fig.4a) that the authors present makes a lot of sense but not all results are consistent with the data:

The experimental curves in Fig.4b do not look different for low and high EGF in the sustained regime. Both the deterministic as well as the semi-deterministic model show a big change in the temporal profiles that is not reflected in the experimental profiles. Fig.4e should also show the EGF low&high in the pulsed regime.

In Fig. 4f, I do not see how the semi-deterministic modeling captures the heterogeneity in the sustained NGF response. The average theoretical curve (Fig.4e) does not contain the initial peak in Erk Activity seen in the experimental data (Fig.4b). More important, none of the theoretical profiles for single cells shows a sustained profile for high EGF or a transient for high NGF which are observed experimentally (Fig.1d,e). Also, the semi-deterministic model does not reproduce the sustained Erk profiles (Fig.4e) when stimulating with 10' high-NGF (Fig.2 b). This points at the model being incomplete or EKAR2G not reflecting the actual Erk activity.

Minor points:

I am curious if the low percentage of sustained response seen after EGF stimulation (Fig.1d, e) corresponds to the percentage of differentiated cells after sustained EGF stimulation. The same is also interesting for NGF if the 20-30% of cells that show a transient Erk activation profile (Fig.1d, e) do not differentiate but instead proliferate? The question is really what the cell perceives as transient or sustained Erk response from the 5 clustered profiles shown in fig1f?

The 3'/10' & 3'/20' application profiles for low-EGF and high-NGF result in similar sustained Erk responses (fig.3e). However, the 3'/10' high-NGF application profile results in more cells that differentiate. More enigmatic is that high-EGF 3'/10' Erk activation profile is more transient than low-EGF 3'/10' Erk activation profile, but results in the same differentiated fraction of cells (compare Fig.3e with Fig.6b).

The immunofluorescence data shown in supp. Fig2a does not reflect the quantification. In case of EGF a clear homogenous transient response can be observed versus a sustained response for NGF. Also, the quantification shows on average no sustained response for NGF (as is observed in the blot in Fig1a).

Some references are cited in the wrong journals.

1st Revision - authors' response

19 September 2015

Reviewer #1:

The authors describe a study analysing MAPK signalling by using a microfluidics set up and life-cell single-cell readout of MAPK activity, and by challenging the network using a defined, pulse-like stimulation pattern they can dissect the timescales shaping the dynamics of the network. The whole study is motivated by the paradigmatic differential response in PC12-cell differentiation, where stimulation with the growth factors NGF and EGF yield distinct responses of MAPK activity (prolonged vs. short) and lead to different phenotypes (neuronal outgrowth vs. proliferation).

The authors observe different short term response of EGF and NGF (which have been studied extensively), but more interestingly, they show that long-term adaptation (or desensitization) of the pathway depends strongly on the stimulus pattern. While sustained stimuli and rapidly repeated short stimuli result in lower amplitudes over time, stimuli intermitted by 60 min result in no decay for EGF stimulation, and reduced decay by NGF stimulation. By using a mathematical model they to build a plausible explanation for this effect: MAPK activity is mediated by fast positive (for NGF) and negative (for EGF) feedbacks, which are integrated in a feed-forward loop (converging at RAF). These two feedbacks together with the feed-forward loop generate the transient vs. sustained MAPK activity that distinguishes the different growth factors. Furthermore, a slow negative feedback via the induction of a phosphatase of the DUSP family causes the adaptive behaviour of the pathway on the time-scale of one hour. Finally, they show that with repeated 3min pulses of EGF (with braks larger then 10 min) they can trigger neurite outgrowth as the ERK kinetics is similar to NGF stimulation.

Overall, I think this is an excellent paper, is well written and such work of challenging the pathway with complex input pattern is timely and had to be done as it unveils important characteristics of the pathway's timescales and feedbacks. While I am no expert in life-cell imaging, I think the work has been well done and the data seems solid. The model seems to be well justified.

While the model plausibly explains the data, it was not really challenged and used to make predictions that could and should be tested experimentally. The prediction that neurite outgrowth can be triggered by pulsed EGF due to less adaptation/dampening at later time points is not really a model prediction but can be seen from the data. Therefore, I think having testable predictions from the model would strengthen the paper further. One of the main or instance, the main desensitizer in the model is the transcriptional induction of DUSP. As DUSP activity acts at the level of ERK, measuring (even only at a single or few time points for example by IF) ERK and MEK could be valuable to distinguish DUSP feedback and other (possibly slow) post-translational feedbacks. Also,

adding a translation inhibitor may be a challenge to dissect the involvement of the slow DUSP feedback, although translation inhibitors may have more unspecific effects.

We are grateful to the reviewer for his positive comments. The reviewer is right that the ability to rationalize GF pulse regimes that optimize ERK activity over time, which then allows us to rewire fate decisions independently of growth factor identity can be seen from the data. We however still believe that using the model to explore which pulse/pause regimes optimize duration of ERK activity is a powerful way to illustrate at which specific frequency we have to stimulate the system to manipulate cell fate.

To challenge our model experimentally, we tested the effect of both transcriptional (actinomycin D) as well as translational inhibitors (cycloheximide) to potentially test for the effect of transcriptionally induced DUSPs. Unfortunately, these drug treatments proved very toxic and led to rapid cell death, that were exacerbated in presence of light-induced phototoxicity, precluding us to perform any accurate measurement on the timescale of multiple hours, which is required to capture DUSP regulation.

The pMEK/pERK experiment suggested by the reviewer is difficult, because we need to induce robust, population homogeneous ERK activity responses to observe this. It is important to understand that while live cell imaging experiments can be routinely performed within the microfluidic device, immunofluorescence experiments are difficult to implement within these devices for two reasons:

- 1. High surface to volume ratio within the 100 um high cell culturing chamber in the microfluidic device makes immunofluorescence studies non-quantitative, most likely because of adsorption of antibodies (we therefore performed the phosphoERK experiments in conventional dishes). This type of study is however suitable for staining with dapi and tubulin antibodies for evaluation of neurite outgrowth (which only requires sufficient SNR to segment cell shape). However, in this setup reproducible measurements of pERK/pMEK fluorescence intensities would be difficult.*
- 2. At this stage it is impossible to fix the cells accurately and rapidly at a certain time point in the microfluidic device. This would require the development of a new device with additional inlets, and very tight control of the delivery of a fixative.*

For now, we therefore cannot offer any perturbation experiments that challenge model predictions. We leave this for future studies in which we will systematically perturb signaling components by RNAi, with the aim to more precisely identify the specific molecular components at play in our signaling network.

Minor: "NGF" is missing after 50 ng/ml in the legend of last figure .

This has been corrected.

Reviewer #2:

Ryu et al. use a combination of FRET reporter, microfluidics, and computational modeling to investigate the impact of ERK/MAPK signaling heterogeneity in PC-12 cells. This question is an important one, as the PC-12 system is a widely used example of how differences in the kinetics of ERK activity can result in different cell fates, but has still not been the subject of careful live-cell study of ERK kinetics. The authors describe substantial heterogeneity in response to both NGF and EGF, then go on to demonstrate that this heterogeneity can be attenuated by pulsed growth factor stimulation. After a thorough characterization of the pulse response kinetics for both growth factors, they construct a mathematical model, using this to understand the potential mechanisms underlying the different kinetics and to predict stimulation patterns that alter the differentiation response to EGF. These predicted patterns are then tested and shown to increase the differentiation response to EGF.

Overall, this study addresses an important open question in the field, is elegantly and rigorously executed, and is beautifully presented. I found this study to be of great interest, and believe it will be a very valuable contribution.

There are a few issues that can be addressed by text or figure changes:

1) The interpretation of Fig. S3a as showing an "all or none" threshold between 0.5 and 1 ng/ml EGF or 1 and 2ng/ml NGF is a bit of a stretch. Certainly the dosage effect appears to be non-linear, but there is clearly some stimulation at 0.5 EGF or 1 NGF visible in the averages. The description of these data should be more in line with what is shown in the figure. Showing some single cell trajectories (or a heatmap) here would be illuminating as to whether individual cells are showing a moderate response at lower EGF concentrations, or whether a few cells are responding with full pulses.

We realized that we made a mistake with data normalization which slightly skewed the curves. We now show the correct data in Fig EV2, and as requested by the reviewer, we also provide representative ERK activity trajectories. We show that there is already substantial 1st peak ERK activity at 0.5 ng/ml EGF and 1 ng/ml NGF, although not at the level of the peak amplitudes observed at 1 and 25 ng/ml EGF, and 2 and 50 ng/ml NGF. These results indicate a dose response curve with a clear non-linear effect, with fixed 1st peak amplitudes above 1 ng/ml EGF and 2 ng/ml NGF. We have downplayed the concept of an all or nothing response at a threshold GF concentration.

2) There is very little discussion of how the reporter, a substrate of ERK, relates to active ERK itself. In the model, these two species are directly equated. However, it is not necessarily the case that these quantities are linearly related (PMID 22114702). Other studies have found it necessary to explicitly model the reporter kinetics (PMID 24848561). It would be useful to justify the details of how this relationship is treated in fitting the model to the data.

We thank the reviewer for this point, we indeed did not explicitly mention this important fact. We have now explicitly emphasized that the EKAR2G FRET probe does not provide an "absolute" measurement of ERK activity, with a necessary "linear relationship" between the FRET emission ratio and ERK activity, and referred to the papers the reviewer suggested. Our immunofluorescence data suggests that the ERK activity kinetics measured by the biosensor are similar to those measured by immunofluorescence (e.g. a steep rise in ERK activity to shape the 1st peak, followed by a slower decrease). An exception to the immunofluorescence-biosensor correspondence is the desensitization kinetics of the 1st ERK activity peak. The EKAR2G emission ratio signal decays slower than the signal measured by the phosphoERK antibody. Arguably, the slower decay is due to biosensor expression affecting the signaling system rather than the biosensor readout itself (Fig. EV1D). We have then clearly stated that we make the reductionist assumption that the FRET emission ratio scales with ERK activity.

Minor issues

Fig. 1b is referenced out of order in the text.
"NGF" missing from the legend in Fig. 6

We have fixed these issues.

Reviewer #3:

This single cell analysis of Erk response to EGF or NGF application regimes is an interesting work that nicely builds on previous knowledge on how distinct temporal Erk profiles that determine cell fate come about. For me the most interesting part is how receptor state affects via feed forward mechanism the positive feedback and negative feedback in the system. This makes the system dependent on the time of growth factor application, which allows the authors the manipulation of cell fate decisions by using pulsed growth factor application regimes. The feed forward mechanism that affects the positive feedback upon NGF stimulation has been previously described in Santos et al. (2007) as being dependent on PKC activity (To affect RKIP) and should be discussed in the text. The molecular mechanism of the feed forward that affects the negative feedback is new (and exciting) and is not discussed in terms of molecular mechanism. One point in relation to this feedback is that stimulation with a low dose of EGF should cause a sustained response because the negative feedback is not activated. Why does this only work in the pulsed EGF application regime? The authors should show the Erk response profiles for the sustained low EGF and NGF dose in Fig.1.

We thank the reviewer for his positive comments. We have now included the PKC/RKIP/positive feedback in the discussion, as a potential feed forward mechanism to modulate the positive feedback. We also have included different potential scenarios as to other signaling networks that might modulate the negative feedback.

We now show representative curves of sustained EGF and NGF in response to sustained low GF dosages in Fig 1G,I. With regard to the negative feedback loop, it is still present even when a low dose of EGF is used. This is evident in multiple experiments: 1. Pulsed EGF stimulation at both low and high dosage leads to adaptation. 2. In response to sustained EGF stimulation, the hint that the strength of the negative feedback loop depends on EGF dosage comes from studying the distribution of the single cell trajectories. One can see in the current Figure 2A-C that a low EGF dose will lead to a higher number of trajectories with a wider tail, suggestive of lower negative feedback, than in the presence of a high EGF dose. This behavior is also captured in our modeling studies (see Figure 5). Thus, even at low EGF dosage, negative feedback dominates, but its strength can modulate the distribution of adaptive and more sustained ERK responses to some extent. In the pulsed EGF application regimes, the EGF-induced ERK activity pulses are adaptive in essence, but simply re-triggered, and the hour-scale negative feedback then gradually reduces ERK activity amplitude, by a 2nd, most likely DUSP-mediated feedback loop.

The other slight concern is that the results heavily rely on a substrate sensor of Erk activity (EKAR2G) whose distribution in the cell as well as its expression level might affect the interaction with phosphatases resulting in different inactivation kinetics. In other words: Erk activity is convoluted with the kinetics of EKARG2 phosphorylation/dephosphorylation that depend on its expression level and partitioning in the cell. The heterogeneity for example in the NGF response could arise from different dephosphorylation kinetics of EKAR2G in the nucleus versus cytoplasm. Can the authors distinguish the responses in these two cell compartments? Also, the fluorescence distribution of the sensor should be provided with the ratiometric images to judge its distribution in the cytoplasm versus nucleus. Even if the sensor rapidly exchanges, EKAR2G in the nuclear volume will dominate the signal. The way the ratiometric images look, one gets the impression that EKARG2 is actually excluded from the nucleus (I would also suggest to make Fig.1c larger to better see the responses). Is there a correlation between EKARG2 expression level and response?

*It looks like we have not sufficiently emphasized that the EKAR2G biosensor we used indeed localizes to the cytosol by virtue of a NES that was also engineered in the 1st generation EKAR sensor published by the Svoboda lab. We have clearly mentioned it: "To study ERK activation dynamics in single PC-12 cells, we produced a stable cell line that expresses EKAR2G, a fluorescence resonance energy transfer-based biosensor for endogenous **cytosolic** ERK activity (Supplementary Fig.1a)". We have now further emphasized that this was made by engineering of a NES in the biosensor in the text. Furthermore, as suggested by the reviewer, we have now produced a panel in which we show both the emission ratio and the subcellular localization of EKAR2G, which is clearly cytosolic (Fig.1B). This therefore excludes the possibility that heterogeneous ERK activity trajectories occur because of different de-activation kinetics in the cytosol and in the nucleus.*

We have not yet explored the dynamics of a nuclear-localized EKAR2G sensor (this is due to the fact that making PC-12 stable cell lines literally takes multiple months). However, as mentioned in the text, previous work from the Zhang lab has shown that there is no time lag between the ERK activation in the cytosol and in the nucleus of PC-12 cells, suggesting that both ERK activity pools are in equilibrium (Herbst KJ, Allen MD, Zhang J (2011) Spatiotemporally regulated protein kinase A activity is a critical regulator of growth factor-stimulated extracellular signal-regulated kinase signaling in PC12 cells. Molecular and cellular biology 31: 4063-4075).

We have shown in Appendix Fig. 1A that the standard deviation of EKAR2G expression is about 50% around mean fluorescence intensity. This is much less than the typically 10-fold difference in expression level observed during transient transfection. We now have additionally correlated ERK activity levels at 40' after NGF stimulation, a time point at which we observe a large amount of signaling heterogeneity with biosensor expression levels (approximated by the donor channel). Our regression analysis shows that both variables are not correlated (Appendix Fig.1E). We would also like to mention that our GF pulse experiments allow us to produce population homogeneous ERK

activity responses, suggesting that heterogeneous ERK activation dynamics are a specific feature that results from sustained GF signaling, but are not the result of a bias introduced by different biosensor expression levels.

The model (Fig.4a) that the authors present makes a lot of sense but not all results are consistent with the data:

We do not claim that our model is absolutely complete. It captures features of the signaling network at specific levels of detail. By example, it is very plausible that receptor-mediated endocytosis will at a certain point kick on the time scale of tens of minutes after GF stimulation, and affect ERK activity dynamics through regulation EGFR/EGF dissociation to some extent. This has not been explicitly modeled because it is not required to explain our data.

The experimental curves in Fig.4b do not look different for low and high EGF in the sustained regime. Both the deterministic as well as the semi-deterministic model show a big change in the temporal profiles that is not reflected in the experimental profiles. Fig.4e should also show the EGF low&high in the pulsed regime.

This discrepancy arises from the fact that the average of the population behavior cannot capture subtle patterns in population signaling heterogeneity observed in response to sustained EGF high and low dosages. The difference between low and high EGF dosages can only be seen clearly using classification of representative ERK activity trajectories, for example by k-means clustering (in now current Fig.2A-C). This allows us to show a shift in the distribution of prototypical ERK activity trajectories towards more sustained responses in the low EGF case. Unfortunately, population averaging obscures this shift in our single-cell data (maybe also confounded by noise and sampling). However, albeit barely visible in the averaged trajectory data, the model still captures this effect when simulated with one fixed parameter set. This explains the apparent discrepancy. We now show all the requested panels in what is now Fig.5E (see below).

In Fig. 4f, I do not see how the semi-deterministic modeling captures the heterogeneity in the sustained NGF response. The average theoretical curve (Fig.4e) does not contain the initial peak in Erk Activity seen in the experimental data (Fig.4b). More important, none of the theoretical profiles for single cells shows a sustained profile for high EGF or a transient for high NGF which are observed experimentally (Fig.1d,e). Also, the semi-deterministic model does not reproduce the sustained Erk profiles (Fig.4e) when stimulating with 10' high-NGF (Fig.2 b). This points at the model being incomplete or EKAR2G not reflecting the actual Erk activity.

The reviewer is indeed right to argue that the simulated ERK activity cell trajectories have a qualitatively slightly different shape than the experimental data. As the reviewer mentions, we did not observe in our simulated curves, 5-10' after stimulation, a marked decline that comes as a "kink". Our simulated curves rather displayed a linear decline over 60'. To address this, we have re-parametrized the model, by additionally simulating heterogeneous expression level of phosphatases (in addition to the signaling components shown in Fig. 5D), as well as further increased the variation of the distributed parameters. The semi-deterministic modeling now captures the experimental trajectories much more faithfully than with the previous parameter set. And as already mentioned above, we now show all the requested panels in Fig.5E.

We believe that our model, simulated with the newly described parameter ranges adequately captures the experimental responses. Our model adequately captures adaptation observed for all EGF/NGF pulses (except for the 10' NGF pulse). As expected, It also shows that the 10' high NGF dosage pulse leads to a mix of sustained/adaptive responses. It faithfully recapitulate the sustained, and the multipulse GF stimulation experiments (using deterministic modeling for the latter). The model, that used the parameter range and signaling components presented in the first version of this paper, captured all these behaviors, but was less faithful in reproducing the precise shapes of the curves.

Minor points:

I am curious if the low percentage of sustained response seen after EGF stimulation (Fig.1, e) corresponds to the percentage of differentiated cells after sustained EGF stimulation. The same is also interesting for NGF if the 20-30% of cells that show a transient Erk activation profile (Fig.1d,

e) do not differentiate but instead proliferate? The question is really what the cell perceives as transient or sustained Erk response from the 5 clustered profiles shown in fig1f?

This question is actually difficult to tackle experimentally. As we mention in the paper, long term imaging of the EKAR2G biosensor results in phototoxicity, detrimental to cell health over the 36-hours timescale required for reading out a robust differentiation cell fate. We therefore cannot correlate a specific, dynamic ERK activity trajectory with a fate (such as appearance of neuronal processes/quiescence versus cell cycle entry). This is the reason why we performed our multipulse differentiation experiments in an incubator with neurite outgrowth in fixed cells as a fate readout. However, part of the answer to the reviewer's question comes from a paper from Tobias Meyer's lab: Chen J-Y, Lin J-R, Cimprich KA, Meyer T (2012) A two-dimensional ERK-AKT signaling code for an NGF-triggered cell-fate decision. Molecular cell 45: 196-209. In this paper it is shown that upon NGF stimulation most of the PC-12 cells will differentiate, but a significant subset (e.g. 20 %) still proliferate. Likewise, a smaller portion of PC-12 cells will remain quiescent when stimulated with EGF (although it was not explored if these cells will differentiate in this paper). Although we cannot yet correlate single cell ERK activity dynamics with fate on a per cell basis, the results from the Meyer lab suggest that heterogeneous ERK activation dynamics can specify different cell fates (e.g. proliferation/differentiation) within a cell population, which was one of the main points of the paper from the Meyer lab. We have clearly discussed this in the paper.

The 3'/10' & 3'/20' application profiles for low-EGF and high-NGF result in similar sustained Erk responses (fig.3e). However, the 3'/10' high-NGF application profile results in more cells that differentiate. More enigmatic is that high-EGF 3'/10' Erk activation profile is more transient than low-EGF 3'/10' Erk activation profile, but results in the same differentiated fraction of cells (compare Fig.3e with Fig.6b).

We agree with the reviewer that the correlation between integrated ERK activity and differentiation is not perfect. One important contribution to this might simply be the sensitivity of our assay. Our data clearly shows that GF pulses, applied at the right frequency to optimize ERK activity duration, can lead to differentiation even with the growth factor EGF. Our assay is sufficiently sensitive to detect that sustained stimulation, or 3'/3' and 3'/60' EGF multipulse regimes, which do not lead to high integrated ERK activity, do not promote differentiation in contrast to 3'/10' and 3'/20' EGF pulse regimes, which optimize for integrated ERK activity. Likewise, our assay shows that the 3'/3' NGF multipulse regime, which induces high ERK activity leads to robust differentiation, while the same 3'/3' EGF multipulse regime, which induces low ERK activity, does not.

The higher integrated ERK activity observed with low versus high 3'/10' EGF multipulse regimes might not exhibit a sufficiently robust increase in the magnitude of integrated ERK activity, as with the other multipulse regimes mentioned above, and therefore elude the sensitivity of our assay. Another important factor might be that not only ERK but also other signaling pathways, that might be triggered differentially by EGF and NGF might be important for differentiation. This might be consistent with the work of Tobias Meyer's lab discussed just above, that shows that cell fate is controlled by both AKT and ERK. Consistently, we also have shown and discussed in the paper, that sustained or 3'/10' multipulse regime low NGF doses, do not lead to robust differentiation as observed with high NGF doses, despite being equally effective at promoting high ERK activity. This suggests that additional, NGF dose-dependent signaling pathways are at work.

The immunofluorescence data shown in supp. Fig2a does not reflect the quantification. In case of EGF a clear homogenous transient response can be observed versus a sustained response for NGF. Also, the quantification shows on average no sustained response for NGF (as is observed in the blot in Fig1a).

We have specifically mentioned in the legend of this figure (now Fig EV1A) that we have slightly saturated the pERK signals at the 5' time point. This is required to visually compare the increased and heterogeneous pERK signals occurring after the first peak in response to sustained NGF versus EGF. Importantly, the pERK western blot shown in Fig.1, as well as the western blots in the PC-12 cell literature in general, are shown as slightly overexposed so as to resolve the sustained (but reduced with respect to the 1st peak) ERK activity. We now provide a statistical test that show that the single cell pERK distributions are significantly different between NGF and EGF after 10'. Furthermore, we provide a density plot that compares phosphoERK distribution within a cell

population at 40' in response to EGF versus NGF. This illustrates that a cell population displays higher ERK activity after the 1st peak in response to NGF when compared to EGF.

Some references are cited in the wrong journals.

We went through the references again, but did not find any problems.

2nd Editorial Decision

15 October 2015

Thank you again for submitting your work to Molecular Systems Biology. We have now heard back from the referee who agreed to evaluate your manuscript. As you will see below, the referee is now satisfied with the modifications made and only lists a few minor issues, which should be addressed in a revision of the work.

REFEREE REPORT

Reviewer #3:

The authors considered and addressed all of the previously raised concerns. The re-parameterization of the model by additional simulation of heterogeneity in phosphatase expression levels now captures much more realistically the experimentally observed behavior and makes the proposed hypothesis more plausible. I have some remaining minor comments and suggestions to improve the paper.

1. The authors basically showed that a low dose of EGF leads to sustained signaling. However, in the new figure 1 the difference between high and low dose of EGF cannot be observed in the average responses due to noise. I believe that their data is actually very consistent over the different experiments, and as shown in Fig. 1G and 2C, in the case of stimulation with low EGF doses, most of the cells (~80%) show a more sustained response, in contrast to the high EGF doses, where the situation is reversed - most of the cells lie below the mean curve and have a transient response. I suggest that the authors apply k-means clustering also to the curves in Fig. 1G and simply present two mean curves reflecting the sustained and the transient populations, together with the number of cells in both cases, which will then be in a nice accordance to Fig. 2C. As a small comment, it is to my opinion not possible to describe an adaptive response based on negative feedback in response to pulsed EGF stimuli as the authors describe in their answers. This is obviously the case because the apparent adaptation follows the EGF profile.

2. Even though the authors state in the manuscript that the differentiation response observed in the pulsed 3'/10' case might be dependent on additional signaling pathways, I suggest that the authors discuss this point in the paper, as they already nicely did in the response letter.

3. The reference of the Santos et al. manuscript is cited in the wrong journal (NCB instead of Nature).

2nd Revision - authors' response

16 October 2015

Reviewer #3:

The authors considered and addressed all of the previously raised concerns. The re-parameterization of the model by additional simulation of heterogeneity in phosphatase expression levels now captures much more realistically the experimentally observed behavior and makes the proposed hypothesis more plausible. I have some remaining minor comments and suggestions to improve the paper.

We thank the reviewer for his positive comments, and suggestions to improve the paper.

1. The authors basically showed that a low dose of EGF leads to sustained signaling. However, in the new figure 1 the difference between high and low dose of EGF cannot be observed in the average responses due to noise. I believe that their data is actually very consistent over the different experiments, and as shown in Fig. 1G and 2C, in the case of stimulation with low EGF doses, most of the cells (~80%) show a more sustained response, in contrast to the high EGF doses, where the situation is reversed - most of the cells lie below the mean curve and have a transient response. I suggest that the authors apply k-means clustering also to the curves in Fig.1G and simply present two mean curves reflecting the sustained and the transient populations, together with the number of cells in both cases, which will then be in a nice accordance to Fig.2C.

We most likely did not sufficiently clearly mention this in the text, but the ERK activity trajectories in Fig.1G and 2 actually are from the same dataset. As the reviewer has rightly understood, picking 10 random ERK activity trajectories from high and low dosages EGF stimulation, and computing their average, does not provide sufficient information to indicate that there are different signaling profiles (Fig.1G). The clustering approach that classifies different signaling profiles shown in Fig.2C is able to pick these subtle changes in population behaviors. So in essence we have already performed the requested analysis suggested by the reviewer. We have modified the text to make this clearer and thank the reviewer for suggesting to clarify this important issue.

The changes are indicated below in bold in the text starting on p.6, 2nd paragraph, continued on p.7:

“To quantify cell heterogeneity, we pooled all high and low dosage, EGF and NGF-triggered ERK activation trajectories **described above (Fig 1E-H)**, and used k-means clustering to extract 5 representative temporal activation patterns (Fig 2A). We then determined their incidence in response to the different GF dosages (Fig 2B,C), and found that EGF favored adaptive responses, while NGF led to a larger number of sustained ERK activity trajectories. We also observed ultrasensitive and switch-like ERK responses for increasing dosages of EGF and NGF. The 1st peak amplitudes saturated and remained unchanged above threshold concentrations of 1 ng/ml EGF and 2 ng/ml NGF (Fig EV2A,B). At lower concentrations, lower and more heterogeneous 1st ERK activity peak amplitudes were observed. A striking observation was that the different representative ERK activity trajectories displayed distinct 1st ERK activity peak amplitudes, that correlated with the ability to produce a transient or a sustained response – cells with low or high 1st peak amplitude respectively produced transient or sustained responses (Fig 2A-D). For additional quantification of this phenomenon, we correlated 1st peak amplitude with long term, 40', ERK activity (indicating transient or sustained ERK activity). Robust correlation was observed for high and low NGF dosages, suggesting that high 1st ERK peak amplitude leads to sustained responses (Appendix Figure 2). Non-significant correlations were observed for high and low EGF dosages. **Another interesting observation was that a low EGF dosage significantly shifted the distribution of ERK activity trajectories towards more sustained profiles when compared to the high EGF dosage (Fig 2A-C). Importantly, such subtle shifts in the distribution of ERK activity profiles are not apparent when a population average is computed (Fig 1G), illustrating the value of our clustering approach to analyze our single cell trajectories datasets.**”

As a small comment, it is to my opinion not possible to describe an adaptive response based on negative feedback in response to pulsed EGF stimuli as the authors describe in their answers. This is obviously the case because the apparent adaptation follows the EGF profile.

All what we want to propose here, is that pulsed growth factor can enable to distinguish between adaptation/bistability as observed with EGF and NGF pulses.

2. Even though the authors state in the manuscript that the differentiation response observed in the pulsed 3'/10' case might be dependent on additional signaling pathways, I suggest that the authors discuss this point in the paper, as they already nicely did in the response letter.

We have clearly mentioned this specific fact in the paper p.14

“This suggests that high NGF dosage is required to trigger additional signaling pathways that are important for neuronal differentiation and/or survival.”

We now have also clearly mentioned this in the discussion part, which then, for sake of clarity, also required us to more precisely discuss our rewiring experiments.

The changes are indicated below in bold in the text starting on p.16, 2nd paragraph:

“Based on our enhanced model of the MAPK signaling network, optimization of integrated ERK activity by delivery of GF pulse regimes at specific frequencies, can then be used to efficiently rewire cell fate decisions independently of GF identity, in the absence of any drugs or gene perturbations. Thus, stimulation with EGF pulses delivered at the required frequency to optimize integrated ERK activity, leads to differentiation rather than proliferation. This might have a number of biotechnological applications. However, we also found that in some instances, GF pulsing schemes that lead to identical integrated ERK activity, do not necessarily lead to the same level of differentiation (Fig 7, high versus low NGF dosages). Not surprisingly, this suggests that depending on the GF, additional signaling networks than ERK are important for cell fate decisions.”

3. The reference of the Santos et al. manuscript is cited in the wrong journal (NCB instead of Nature).

We have fixed this issue.